



# Time Series Analysis of C-Band Sentinel-1 SAR Over Mountainous Snow with Physical Models of Volume and Surface Scattering

Firoz Kanti Borah[1], Jonas-Fredrick Jans[2], Zhenming Huang[1], Leung Tsang[1], Hans Lievens[2], and Edward Kim[3]

[1]Radiation Laboratory, Department of Electrical Engineering and Computer Science, University of Michigan, Ann Arbor, MI, 48109, USA
[2]Hydro-Climate Extremes Lab, Department of Environment, Ghent University, B-9000, Gent, Belgium
[3]NASA Goddard Space Flight Center, Greenbelt, MD, USA

**Correspondence:** Firoz Kanti Borah (firozb@umich.edu)

**Abstract.** In this article, we analyze the time series data collected by the Sentinel-1 C-band synthetic aperture radar over the Alps mountains in Southern France. Both the co-polarized and cross-polarized radar data are analyzed. We study the combined effects of the volume scattering of snow and the rough soil surface scattering below the snow using physical models to show the contributions from both components at C-band. For volume scattering, the bi-continuous DMRT equations are used to obtain backscattering coefficients of the dense media layer. To calculate the rough surface scattering component, 3D Numerical Solution of Maxwell's Equations (NMM3D) method is used. The bi-continuous DMRT model is based on the dense media radiative transfer where the microstructure of the media is controlled by two parameters: mean grain size $< \zeta >$ and aggregation parameter $b$. Radiative transfer (RT) equations are used to obtain backscattering coefficients of the dense media layer. In the dense media layer, we consider ice grains aggregated into clusters. The NMM3D model is a method of moments (MoM) based electromagnetic solver that calculates backscattering from a rough surface of a given permittivity, rms height and correlation length. For applications to remote sensing of terrestrial snow at C-band, the results of DMRT simulations show that cross-polarization components of snow volume scattering at large snow depth are larger than that of the of soil surface scattering. The time series are analyzed by varying the snow parameters with time yet keeping the rough surface parameters constant with time. There is good agreement between the time series of the modelled and measured backscatter at both co-pol and cross-pol within 2dB. Results are presented for three seasons between 2017 and 2020 and for four different locations in the Alps. Using this physical model of volume and surface scattering, we also discuss the reason for Sentinel-1's sensitivity in cross pol to snow.



## 1   Introduction

Snow is a critical part of water resource management, and many people rely on snow for water, particularly in the Northern Hemisphere. Thus, it is critical to monitor and map snow cover, as well as accurately retrieve snow water equivalent (SWE) in mountain regions. In the past, many researchers have used passive microwave remote sensing (radiometers) to monitor snow cover and SWE such as Ulaby and Stiles (1980), Chang et al. (1982), Foster et al. (2005), Vuyovich et al. (2014) etc. The launch of the AMSR-E radiometer satellite in 2002 also resulted in more studies on passive microwave remote sensing of snow

and SWE retrieval (Kelly et al. (2003)). Radiometer brightness temperature measurements show very good sensitivity to snow depth (Foster et al. (2005), Lemmetyinen et al. (2018)), but only over a very large spatial footprint, typically about tens of kilometers. AMSR-E can measure horizontally and vertically polarized brightness temperatures at 6.9 GHz, 10.7 GHz, 18.7 GHz, 23.8 GHz, 36.5 GHz, and 89.0 GHz but the spatial resolution of the individual measurements varies from 5.4 km at 89 GHz to 56 km at 6.9 GHz (Lobl (2001)). Such low resolutions limit the capability to observe the spatial variability of the

snow. This becomes an issue as the snow packs can vary drastically from one location to another within a few hundred meters, especially in mountainous regions. With a spatial resolution in the kilometer scale, it is also difficult to distinguish between emissions from a snow pack in an open ground and emissions from a snow pack covered by vegetation or forests.

To mitigate the issues with passive remote sensing, scientists have shifted focus to active remote sensing. In the past studies

on active measurements of snow were largely theoretical, such as Tsang and Kong (1992), Mätzler (1998), as there were not enough active snow measurements. With the advent of synthetic aperture radar (SAR) technology, particularly airborne SAR systems, researchers have characterized the backscattering of snow at multiple frequencies from C-band (5.4 GHz) up to X- and Ku- band of 17.2 GHz (Shi (2006), Yueh et al. (2008)) at great extent and successfully performed SWE retrievals from the active backscattering measurements at X and Ku bands (Zhu et al. (2018), Borah et al. (2023),Durand et al. (2024), Singh

et al. (2024), Pan et al. (2024)). Satellite missions based on X and and Ku band volume scattering have also been proposed (Rott et al. (2010), Derksen et al. (2021), Tsang et al. (2022)). The big advantage of SAR compared to a radiometer is the very high spatial resolution of SAR. With SAR, one can achieve resolution in the order of meters which is crucial in studying spatial variability. Hoppinen et al. (2024) showed that repeat pass interferometric SAR (InSAR) at L band can also be used to monitor the change in snow depths. Lei et al. (2016) showed that single pass, dual antenna InSAR can also be used to retrieve

snow depths as snow depth shows sensitivity to interferometric phase difference at C-band. Leinss et al. (2016) showed that the anisotropy of snow can be esitimated using polarimetric phase difference of radar time series using TerraSAR-X. Although these recent airborne missions have increased our understanding of microwave snow scattering, presently, there is only one dedicated satellite mission on snow, TSMM (Derksen et al. (2021)), that is yet to be launched.

An alternative satellite mission to monitor and map snow globally is Sentinel-1, launched by the European Space Agency (ESA) and Copernicus. It has been shown that the study of Sentinel-1 cross polarization data can provide important retrieval of SWE in mountainous snow (Lievens et al. (2019) ). It is important to understand these physics by using an accurate physical





forward model which in turn can be used to retrieve SWE or snow depth. The C-band backscattering results and retrieval are complementary to the X- and Ku-band satellite missions (Rott et al. (2010), Derksen et al. (2021)) for large snow depth, e.g., of
1 meter and beyond. Recent studies also show how Sentinel-1 in conjecture with reanalysis SWE can aslo be used to monitor the snow cover (Lei et al. (2023)). Lievens et al. (2019) showed that Sentinel-1 cross-pol data can be used to monitor and map snow depth in the Northern Hemisphere. It is because the data show that the C-band Sentinel-1 cross-pol backscatter increases with snow depth. But at the same time, the co-pol signal does not show any sensitivity to the snow depth. This has surprised researchers as it was believed that the volume scattering at C-band to be very small compared to the rough soil surface scattering
below the snow because the wavelengths at C-band (5.5 cm) is much larger than typical snow grain size of 1mm (Nagler and Rott (2000)). Which means that, it was assumed that C-band will not have sensitivity to snow depth. But the results of Lievens et al. (2019) shows that cross-pol signal does have sensitivity to the snow depth and at the same time the co-pol signal does not. This sensitivity of the cross-pol signal was shown to have potential to monitor deep mountain snow globally at C-band. This strong cross-pol along with insensitive co-pol has been a conundrum to solve. Some physical reasoning and explanation
for this may be anisotropy of the ice crystal, dense ice crystal aggregates in the deep mountain snow or inhomogeneity in the snow pack. In this paper, we explain the strong cross-pol sensitivity to snow accumulation in mountainous regions by dense aggregates of ice clusters combined with the effect of rough soil surface scattering below the snow pack. For forward modelling, this paper uses the bi-continuous DMRT model with dense aggregates of ice grain clusters to calculate the volume scattering from the snowpack and NMM3D model to simulate the rough surface scattering.


For volume scattering contributions, the bi-continuous DMRT model has been used in this paper. Multiple scattering models have been proposed to compute the effective permittivity (Siqueira and Sarabandi (2000)) and scattering (Tsang and Kong (1992), Mätzler (1998), Tan et al. (2015)) for dense random media (which is used for characterizing snow). There are also semi-empirical models to calculate the back scattering from snow such as sRT (Rott et al. (2010)), and MEMLS3&a (Proksch
et al. (2015)), but these models rely on an empirical fit of a snow microstrurcture parameter using ground measurements to reduce complexity and increase computation speed. This results in inaccurate backscatter predictions especially when the empirical fit may not characterize the dense media and dense aggregates accurately. The dense media radiative transfer model (DMRT) is based on near field wave interactions based on the phase matrix of the dense media and far field intensity interaction based on radiative transfer equation. In QCA/DMRT these interactions are found using the sticky sphere model as random
media. These densely packed sticky spheres are solved for scattering using the QCA/DMRT (Tsang and Kong (1992)). The scattering of sticky sphere model has also been studied in improved born approximation (IBA) (Mätzler (1998)) and both these models predict a small cross-polarization because of the use of spherical shaped particles. In recent years, community based Snow Microwave Radiative Transfer (SMRT) models (Picard et al. (2018), Sandells et al. (2021)) have been published based on synthesis of various snow microwave scattering models. Although many people have used spheres to model ice in snow,
ice crystals, particularly when the ice aggregates, have very irregular shapes. As we are dealing with very deep snow, the ice crystals in the snow pack can very easily form dense ice aggregates as hypothesized in Lievens et al. (2019), King et al. (2015), which are highly irregular in shape and which can explain the strong cross-pol. The merits of bi-continuous DMRT





are that (i) the computer-generated microstructures resemble real snow, (ii) it gives lower frequency power law dependence than most models, in particular the classical Rayleigh scattering, which has frequency dependence to the $4^{th}$ power, (iii) gives strong cross-polarization because of irregular shapes aggregates and (iv) full wave simulations of 3D Maxwell equations gives accurate calculations of the phase matrix and the cross-polarization component of phase matrix and gives includes near field interactions among the aggregates. This makes bi-continuous DMRT model suitable for this problem as the snow in bi-continuous DMRT model can be represented as densely packed irregular shaped ice crystals (Ding et al. (2010),Xu et al. (2012),Xiong et al. (2022)). The full wave simulations aspects of bi-continuous DMRT are also different from prior models that have been based on analytical approximations. Full wave simulations give accurate cross polarization results which are difficult to obtain for analytical approximation.

Thus, to date, the bi-continuous DMRT model (Ding et al. (2010), Xu et al. (2012), Xiong et al. (2022), Zhu et al. (2023)) has been most successful in predicting the large sensitivity of cross-polarization to changes in snow depth. Cross-polarization observations are used in this paper because they are the basis of snow depth retrieval at C-band (Lievens et al. (2019)). The bi-continuous model is based on computer generated snow microstructures followed by exact solutions of 3D Maxwell equations of full wave simulations. The full wave simulations aspects are also different to prior models that have been based on analytical approximations. A theoretical basis of large cross-polarization has been proposed by Zhu et al. (2023) while in this paper, we use this theoretical basis to match the time series data of Sentinel-1.

The rough surface component of the physical model is governed by the NMM3D model which is characterized by the relative soil permittivity between the ground and the media above it, root mean squared height of the surface and the correlation length of the surface. For the study of backscattering from a bare surface, there are empirical, analytical, and numerical models. For analytical models, the small perturbation method (SPM) (Rice (1951)), small slope approximation (SSA) (Voronovich (1994)), and advanced integral equation method (AIEM) (Wu et al. (2008)) are well-known and applied. But these analytical models contains approximations which make them difficult to use in different varying physical situations. For example, the SPM method is only applicable for surfaces with very small roughness ($kh < 0.3$, where $k$ is the wave number ($2\pi/\lambda$) and $h$ is the rms height). Empirical model, such as Oh's model (Oh et al. (2002)) is also extensively used. This model is based on empirical fit of measured data using a few points and does not cover an extended range of physical rough surface parameters like permittivity or rms height. In order to have extended physical parameter coverage and accurate results, one needs to use numerical methods. There have been many numerical methods proposed to calculate scattering from rough surfaces, such as the method of moment (MoM) (Johnson et al. (1996), Liao et al. (2016)), extended boundary condition method (Duan and Moghaddam (2012)), and finite-element method (Lawrence et al. (2011)). In this paper, we use the 3-D numerical method of Maxwell equation (NMM3D), which is a 3-D full wave solution of Maxwell's equations for rough surfaces using Monte Carlo simulations. NMM3D was developed with a sparse matrix canonical grid (SMCG) method on the MoM surface integral equation(Tsang et al. (2001),Huang et al. (2010),Qiao et al. (2018)). Near field preconditioning (Liao et al. (2016)) was also introduced to enhance convergence in iteration method. This makes NMM3D with SMCG and near field preconditioning much





faster than traditional numerical methods. Liao et al. (2016) showed that NMM3D were shown in good agreement with ground-based measurement for C-band soil surface application for both co polarization and cross-polarization. The NMM3D results

at C band for this present paper show that the rough surface co-polarization is stronger than co-polarization of snow volume scattering but is weaker than cross polarization of volume scattering at large snow depth. This combination of bi-continuous DMRT for volume scattering and the NMM3D for rough surface scattering is used in this paper.

The organization of the paper is as follows. In section 2, the data collected by the Sentinel-1 over the Alps and the trend that

the data are described . In section 3 the volume and surface scattering forward model and the effects of the two contributions in C-band are described. In section 4, the analysis of time series data of Sentinel-1 the one season from a single location are first carried out with the forward model. We explain the physics of both co- and cross-polarization as seen by the Sentinel-1. We next extend the time series analysis to several more locations with multiple seasons. Section 5 is the conclusion of the paper.

## 2 Sentinel-1 Dataset

Although Sentinel-1 has an extensive coverage over the Alps mountain regions in the Northern Hemisphere, we selected specific sites to analyze as these sites had measurements of snow water equivalent. We selected data collected over locations without any vegetation so that the only scattering contributions that the Sentinel-1 radar sees are arising from the volume scattering in snow and the rough soil surface scattering below the snow layer. The backscatter data, processed following Lievens et al. (2022), for three of such locations are plotted in Figures 1, 2 and 3 which correspond to $(44^0 \ 40^{'} \ 12^{''}$ N, $6^0 \ 25^{'} \ 12^{''}$ E),

$(45^0 \ 17^{'}24.72^{''}$ N, $6^0 \ 39^{'}33.12^{''}$ E) and $(44^0 \ 40^{'}30.36^{''}$ N, $6^0 \ 56^{'}48.84^{''}$ E) respectively. The local incidence angle (LIA) of the signal at the locations is available from the satellite data.

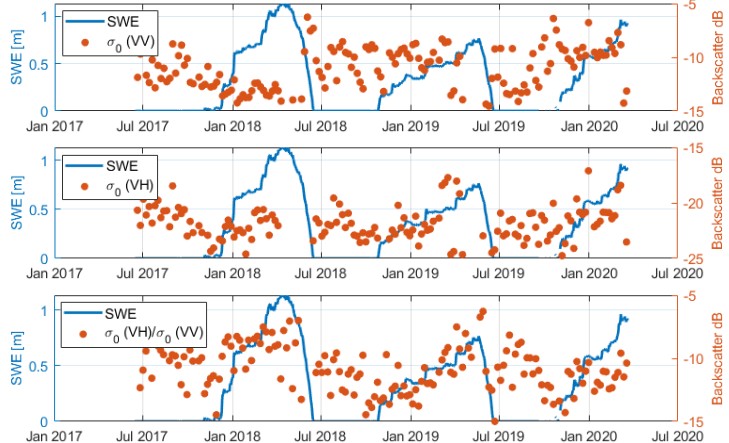

**Figure 1.** Backscatter and SWE time series at Alps location 1 $(44^0 \ 40^{'} \ 12^{''}$ N, $6^0 \ 25^{'} \ 12^{''}$ E). The top, middle and bottom figure correspond to the co-pol $\sigma_0^{VV}$, cross-pol $\sigma_0^{VH}$ and the ratio of cross- to co-pol $(\sigma_0^{VH}/\sigma_0^{VV})$ respectively.



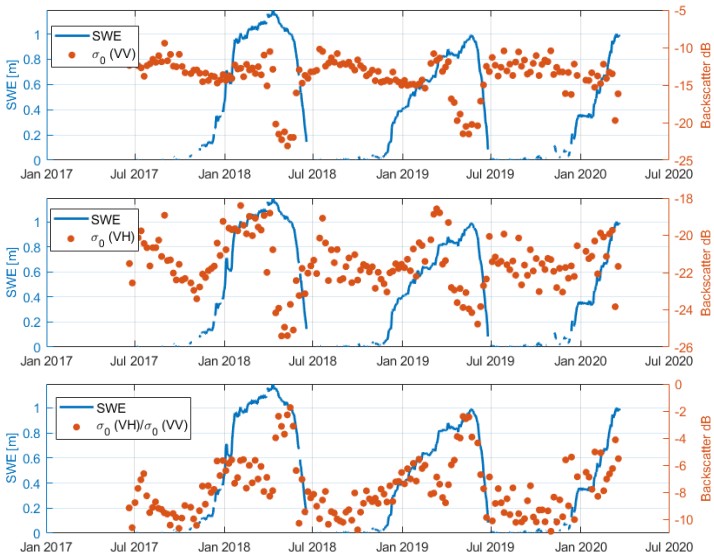

**Figure 2.** Backscatter and SWE time series at Alps location 2 ($45^0\ 17'\ 24.72''$ N, $6^0\ 39'\ 33.12''$ E). The top, middle and bottom figure correspond to the co-pol $\sigma_0^{VV}$, cross-pol $\sigma_0^{VH}$ and the ratio of cross- to co-pol ($\sigma_0^{VH}/\sigma_0^{VV}$) respectively.

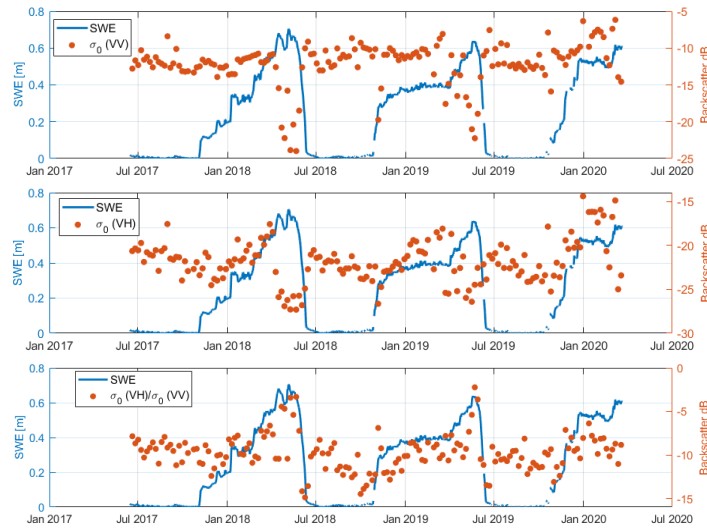

**Figure 3.** Backscatter and SWE time series at Alps location 3 ($44^0\ 40'\ 30.36''$ N, $6^0\ 56'\ 48.84''$ E). The top, middle and bottom figure correspond to the co-pol $\sigma_0^{VV}$, cross-pol $\sigma_0^{VH}$ and the ratio of cross- to co-pol ($\sigma_0^{VH}/\sigma_0^{VV}$) respectively.





All three locations are plotted from summer of 2017 (July 2017) till end of winter of 2020 (March 2020). The snow depth in these regions consistently cross more than 2 m (SWE more than 0.6 m). During the three snowfall seasons in location 1,

we can see that the co-pol is around -13 dB for the 2017-18 season and around -10 dB for both the 2018-19 and the 2019-20 season. The cross-pol on the other hand increases from around -25 dB to -20 dB for the 2017-18 season and increases from around -23 dB to -18 dB for the 2018-19 and the 20119-20 season. This trend is also reflected in the ratio of cross-pol to co-pol. The ratio also increases as the cross-pol increases with snow but the co-pol remains relatively same. The same trends can be seen in the other two locations where the co-pol is insensitive to the snow depth but the cross-pol increases as the snow depth

increases. Similar trends are also reported in Lievens et al. (2019) which has data from different locations as well. The question of why co-pol has no sensitivity but the cross-pol has 5 to 6 dB of sensitivity was raised in Lievens et al. (2019) and was tried to answered in Zhu et al. (2023). In the next sections we show in detail how the combination of contributions from the volume scattering and the rough soil surface scattering leads to this effect of different sensitivities in co- and cross- pol. Please note that in this paper, 100m resolution of Sentinel-1 SAR data is used so that the fixed locations used have as little spatial variability

as possible. The result of that choice is that the data sometimes has a very high variability from a day to day value. We are focusing on explaining the trend of the data rather than trying to explain the individual points in the time series.

## 3 Forward Modelling at C-band of Volume Scattering of Snow and Surface Scattering of Rough Soil Surface

The Sentinel-1 radar measures the total backscatter of the illuminated area. This means that it is measuring both the volume scattering (the scattering from the snow pack itself) and the rough soil surface scattering. The total scattering contribution from

both the snow volume and rough surface can be given by

$$\sigma^{tot} = \sigma^{vol} + \sigma^{rs} e^{-2\tau \sec \theta_t} \tag{1}$$

In 1, $\sigma^{vol}$ is the volume scattering calculated from the bi-continuous DMRT model. The bi-continuous DMRT is solved numerically so that $\sigma^{vol}$ includes volume multiple scattering effects. The quantity $\sigma^{rs}$ is the surface scattering of rough soil surface from calculated from the NMM3D model. The $e^{-2\tau \sec \theta_t}$ term is the attenuation of the wave travelling through the snow pack.

It depends on the optical depth $\tau$ of the snow pack, which can be found from the bi-continuous media DMRT model, and the transmitted angle $\theta_t$ can be found from the Snell's law at the air snow interface. This is shown in Fig. 4.

### 3.1 Volume Scattering

The bi-continuous DMRT model is used to calculate the volume scattering from the snow. In the bi-continuous DMRT model, first a bi-continuous two-phase random medium that consists of ice particles with a relative permittivity of 3.2 and air back-

ground is generated to represent the snow pack. The physical characterization of the snow is controlled by two parameters, $<\zeta>$ and $b$ which correspond to the inverse of mean optical grain size and aggregation of the ice crystals, respectively. The smaller the value of the parameter $b$ is, the larger and more denser are the aggregates. Once the bi-continuous media has been generated, the phase matrix of the bi-continuous media is calculated using full wave simulations of Maxwell's equations. The



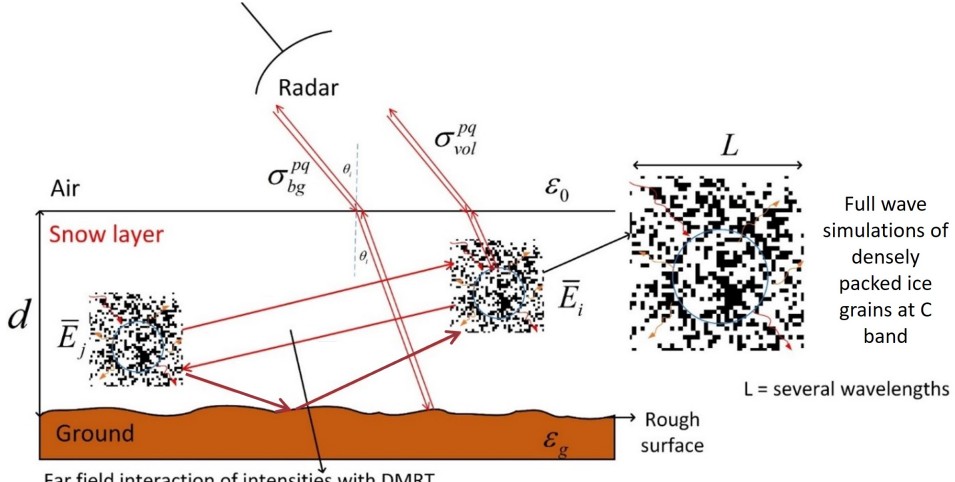

**Figure 4.** Volume and surface scattering interactions as seen by a radar. The volume contribution $\sigma_{vol}$ comes from the near and far field interactions of the densely packed ice clusters and the surface contribution $\sigma_{bg}$ comes from the bare rough surface scattering $\sigma_{rs}$ attenuated by the snow on top.

phase matrix is then used in the DMRT equations to predict the backscatter from snow based on inverse mean grain size

($< \zeta >$), aggregation $b$, snow depth, snow density, and the incidence angle of the incoming wave. The bi-continuous media model was proposed in Ding et al. (2010).

As shown by the Sentinel-1 data, C-band in the mountainous snow has a higher sensitivity to cross-pol but very little sensitivity to co-pol. This is due to a combination of rough surface scattering and dense aggregation in the snow pack which

can occur in the very deep mountainous snow which in shown by. Rough surface scattering will be discussed in detail in the next section. The random media generation of the snow pack in the bi-continuous DMRT model is based on a random position function $S(\bar{r})$ given by

$$S(\bar{r}) = \frac{1}{N} \sum_{n=1}^{N} \cos(\bar{\zeta}_n \cdot \bar{r} + \phi_n) \tag{2}$$

where $\bar{\zeta}_n$ is a random wave vector with $\bar{\zeta}_n = \zeta_n \hat{\zeta}_n$. The unit vector $\hat{\zeta}_n$ is uniformly distributed on a unit spherical surface.

The phase $\phi_n$ is a random variable uniformly distributed between 0 and $2\pi$. The magnitude $\zeta_n$ is a random variable, which is given by a probability density function (pdf) $p(\zeta)$ governed by a gamma distribution. There are two parameters,$< \zeta >$ and $b$, in gamma distribution.$< \zeta >$ is the mean value of $\zeta$, which is inversely proportional to grain size with a unit of $m^{-1}$ and $b$ is a dimensionless aggregation parameter and inversely related to the standard deviation of $\zeta$. To generate dense aggregates, parameter $b$ should be less than 1. With a fixed $\langle \zeta \rangle$ and b < 1, the issue is that when $\zeta$ approaches zero, $p(\zeta)$ is approximately

proportional to $(\zeta/ < \zeta >)^b$ and decays slowly. This results in non convergent phase matrix and scattering results (Zhu et al.





(2023)). To solve this problem, $p(\zeta)$ is adjusted to make it decay faster near zero by adding an exponential decay. The bi-continuous media governed by this new $p(\zeta)$ is called the tuned bi-continuous media. The tuned $p(\zeta)$ is

$$
p(\zeta) = \begin{cases} a \exp\left(-\frac{\zeta_j}{\zeta}\right), & \text{for } \zeta \leq \zeta_j \\ \frac{(b+1)^{b+1}}{\Gamma(b+1)<\zeta>}\left(\frac{\zeta}{<\zeta>}\right)^b \exp(-(b+1)\left(\frac{\zeta}{<\zeta>}\right)), & \text{for } \zeta > \zeta_j \end{cases} \tag{3}
$$

In (3), the gamma distribution is truncated at $\zeta_j$ and replaced by an exponential function that has an argument inversely

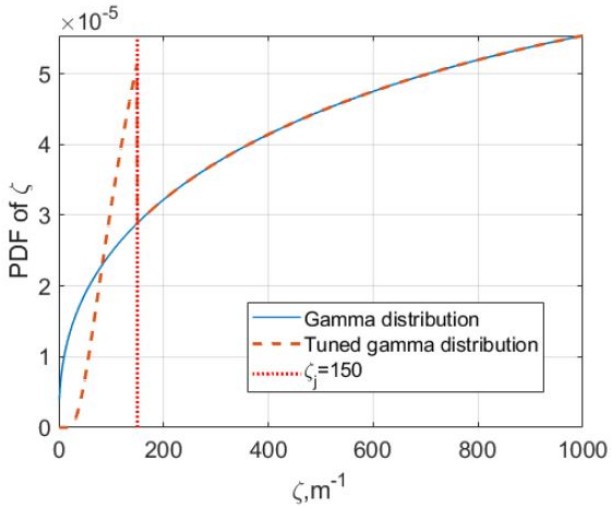

**Figure 5.** Comparison of the pdf of $\zeta$ between the gamma distribution and the tuned gamma distribution for $<\zeta> = 11000 \text{ m}^{-1}$, $b = 0.4$, and the truncated $\zeta_j = 150 \text{ m}^{-1}$.


proportional to $\zeta$. The coefficient $a$ is determined by normalization of the pdf. As shown in Fig 5, the tuned $p(\zeta)$ decays much faster near zero than the regular gamma distribution. Using this tuned bi-continuous media formulation, we can generate a microstructure which resembles snow as given by 6.

Once a volume bi-continuous media has been generated, the next step is to apply Dense Media Radiative Transfer (DMRT) model using discreet dipole approximation (DDA) to find the scattered fields from the snow volume and then the backscattering coefficients. An overview of DMRT's governing processes is given in Fig. 4. Step 1 consists of a generation of volume bi-continuous media. The second step is to apply the full-wave simulations using the discrete dipole approximation (DDA) to compute the compute the scattering, absorption and extinction coefficients, $\kappa_s$, $\kappa_a$ and $\kappa_e$ respectively, and phase matrix $\bar{\bar{P}}$ of

dense aggregates samples with sample sizes of several wavelengths. The third step is to solve RT equations with calculated phase matrices and extinction coefficients to obtain the backscattering coefficient of a layer of snow pack. In the second step



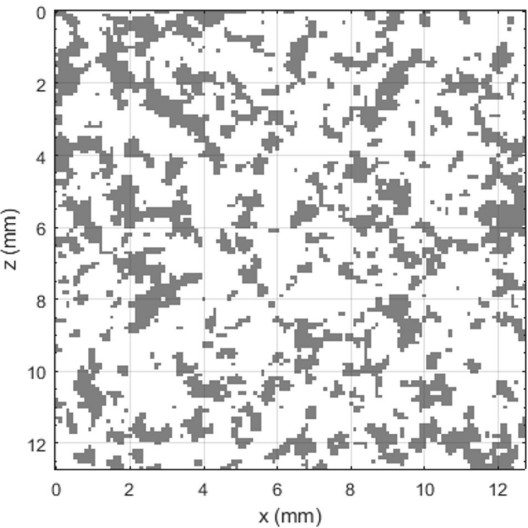

**Figure 6.** Bi-continuous Media generation characterized by $< \zeta > \, = 11000 \; \mathrm{m}^{-1}$, $b = 0.4$, and the truncated $\zeta_j = 150 \; \mathrm{m}^{-1}$.

of using DDA to compute the phase matrix, the sample volume V is discreetized into small cubes. For each realization, DDA equations are solved, and then, scattered far fields of the sample is calculated. Using Monte Carlo simulations, realizations averaging is used to obtain the coherent and incoherent field. The coherent and incoherent bistatic scattering cross sections

are, respectively, the coherent and incoherent scattered wave intensities normalized by the incident wave intensity. The phase matrix P is the bistatic scattering cross section per unit volume. The incoherent phase matrix is used in the RT equations.

The calculated phase matrix, scattering coefficient, and extinction coefficient calculated from sample volumes are substituted into the RT equations.

$$\frac{d\bar{I}(\bar{r}, \hat{s})}{ds} = -\kappa_e \bar{I}(\bar{r}, \hat{s}) + \int_{4\pi} d\Omega' \bar{\bar{P}}(\bar{r}, \hat{s}, \hat{s}') \cdot \bar{I}(\bar{r}, \hat{s}') \tag{4}$$

where I is the specific intensity at the position $\bar{r}$ in direction $\hat{s}$ inside the random media. In the third step, the RT equations are solved by the eigen-quadrature approach. Simulations given in this article consider a single uniform medium layer. For random media with multiple layers, the proposed model is also applicable. Phase matrix and scattering properties are computed for each layer. Then, substitute them into the RT equations to calculate total backscatter of all the layers. The raio of cross polarization

can be simply explained by the intuitive relation-

$$\frac{\text{cross-pol}}{\text{co-pol}} = \frac{(\text{cross-pol of volume scattering + cross of surface scattering})}{(\text{co-pol of volume scattering + co-pol of surface scattering})}$$

Volume scattering increase with snow depth while surface scattering decreases slightly with snow depth. Thus for the cross-pol/co-pol ratio to increase with snow depth, we need to have cross pol of volume to be larger than cross pol of surface scattering while the co-pol of volume scattering to be less than co-pol of surface scattering.





## 3.2 Rough Soil Surface Scattering

The rough soil surface below the snow pack also has a significant impact, particularly at co-pol as rough surface scattering has strong contributions at co-pol even with the attenuation from the snow pack. Therefore, it is important to accurately characterize and calculate $\sigma^{rs}$ from Eq. 1. In this paper, we use 3D Numerical Maxwell's Equations (NMM3D) solver with sparse matrix canonical grid (SMCG) and near field preconditioning. NMM3D is a method of moments (MoM) based numerical method using Monte Carlo simulation. For each realization, a unique height profile $z = f(x,y)$ from a Gaussian random process is generated based on given rms height, correlation length, and correlation function. Fig. 7 shows the surface profile for the exponential correlation function for $256 \times 256$ grid points with surface size of $16 \times 16\lambda^2$.

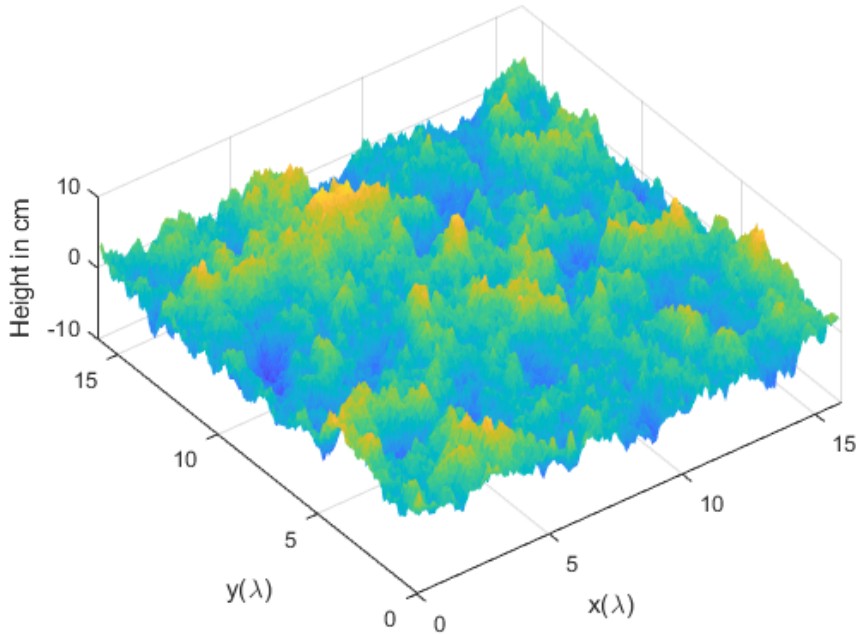

**Figure 7.** Computer generated 3-D profiles for NMM3D simulation (exponential correlation function).

To solve this problem, the well known Poggio–Miller–Chang–Harrington–Wu (PMCHW) formulation is used to get the integral equations needed to solve. These equations are

$$\hat{n} \times (L_1 + L_2)\bar{J}_s - \hat{n} \times (K_1 + K_2)\bar{M}_s \quad = \quad \hat{n} \times \bar{E}_i(\bar{r}) \tag{5}$$

$$\hat{n} \times (K_1 + K_2)\bar{J}_s + \hat{n} \times \left( \frac{1}{\eta_1^2}L_1 + \frac{1}{\eta_2^2}L_2 \right)\bar{M}_s \quad = \quad \hat{n} \times \bar{H}_i(\bar{r}) \tag{6}$$





The $L$ and $K$ operators are defined in Liao et al. (2016). The surface is then discreetized using the Rao–Wilton–Glisson (RWG) basis function and the Galerkin's method to find the surface unknowns $\bar{J}_s$ and $\bar{M}_s$ and then subsequently find the scattered fields. A tapered plane wave is used to avoid diffraction from the edge of a finite-size surface. For the impedance matrix of NMM3D, a different fast method is applied based on interaction of the electric fields. For the near field strong interaction, MoM is applied directly, using near field preconditioning to accelerate the GMRES solver. For far field weak interaction, SMCG is applied by using a fast Fourier transform. The boundaries of these two regions are adjustable to balance the results accuracy and memory efficiency. Based on the tapered incident wave and the calculated scattered wave, the bistatic scattering coefficient is determined by

$$\gamma_{\beta\alpha}(\theta_s, \phi_s, \theta_i, \phi_i) = \lim_{r \to \infty} \frac{4\pi r^2 \mid E_\beta^s \mid^2}{\mid E_\alpha^i \mid^2 A \cos\theta_i} \tag{7}$$

Once we have both volume and surface scattering, we can then calculate contributions from both of the components. The dense aggregate model, with rough surface scattering, shows that the volume scattering of both co-pol and cross-pol is much less than the surface scattering of both co- and cross-pol. As the snow depth increases, the volume scattering at both co-pol and cross-pol increases. The rough surface scattering starts decreasing because of the attenuation from the snow pack. But, when the snow depth is very high the co-pol volume and surface scattering are comparable (which make the total signal insensitive to the snow depth) whereas the cross-pol volume scattering is much higher than the cross-pol rough surface scattering at very high snow depth. This makes cross-pol more sensitive to the snow depth as volume scattering in cross-pol is dominant. This is shown in Fig. 8, where the volume and rough surface scattering (with attenuation) is plotted against snow depth at C-band. The parameters for the random bi-continuous media include volume fraction $f_v = 0.3$, $<\zeta> = 11000 m^{-1}$ (1mm optical grain size) and aggregation parameter $b = 0.4$. The rough surface below the snow is characterized by ground permittivity of 9+1.5i and rms height of 1.5 cm. Table 1 shows that at 200 cm of snow depth, the volume and surface scattering of co-pol are comparable but the volume scattering at cross-pol is 5 dB (3 times) higher than the surface scattering. And as seen in Figures 1, 2 and 3 from section 1, in mountain regions, the snow depth constantly goes beyond 2m. This is the reason Sentinel-1 sees high sensitivity of cross-pol to snow depth.

| | Snow Depth (cm) | Volume Scattering (dB) | Surface Scattering (dB) |
|---|---|---|---|
| Co-pol | 50 | -15.2 | -8.8 |
| VV | 200 | -9.4 | -10 |
| Cross-pol | 50 | -24.3 | -19.1 |
| VH | 200 | -15 | -20.2 |

**Table 1.** Comparison of co-pol and cross-pol backscatter at shallow and very deep snow depths.





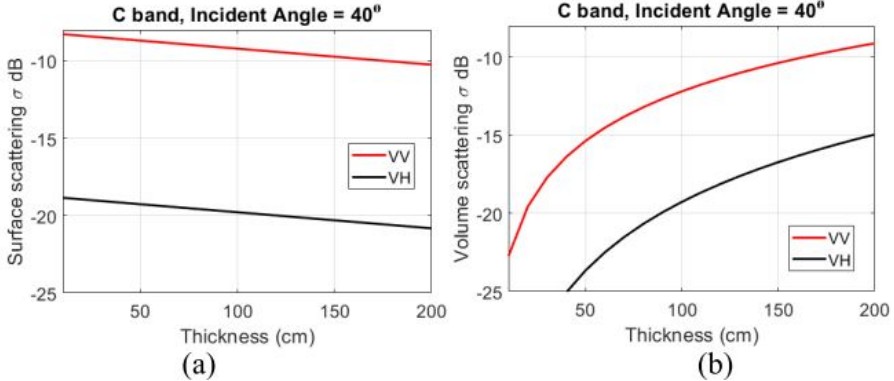

**Figure 8.** (a) Surface scattering and (b) volume scattering as a function of thickness for a layer of dense aggregates at 5.4 GHz. At shallow depths, the rough surface scattering of bo co- and cross-pol are much higher compared to volume scattering. At very high snow depths, the surface and volume scattering becomes comparable in co-pol but the surface scattering is much lower than the volume scattering at cross-pol. This results in higher sensitivity of cross-pol to snow depth at C-band.

## 4    Sentinel-1 Time Series Data Analysis with Volume and Surface Scattering

We next investigate the summer to winter period for a selected location in the Alps. This is shown in Fig. 1, where the measured
data is plotted from July 2017 till March 2020.We will analyze the winter season of 2017-18 for location 1, that is from 15 June 2017 till 21 March 2018. The plot is separated into five points in the time series: 1, 2, 3, 4 and 5. Each of these points represents a change in the snowpack at that time point. Starting from point 1 (Aug 5, 2017), there is no snow (no volume scattering) and only rough soil surface scattering between air/soil interface is present. The cross pol rough surface scattering is about 10 dB below co-pol. On point 2 (Nov 5, 2017), there is a little snowfall, but the volume scattering at C-band due to this
snowfall is very small at both co- and cross-pol. Hence the purple dot (volume scattering) at point 2 is much lower than the black dot (rough surface scattering) and most of the total scattering is from the rough surface with a little bit of attenuation as given by equation 1. On point 3 (Dec 11, 2017) the snow depth has increased as compared to point 2. At co-pol, there is only rough soil surface scattering between snow/soil with little attenuation from snow. However, at cross-pol, there is already some contribution from volume scattering along with rough surface scattering. On point 4 (Jan 04, 2018), both volume and rough
surface contribute to the total scattering at co-pol, but in cross pol, the volume scattering dominates much more than the rough surface. Finally, during the peak snowfall in point 5 (Mar 06, 2018) volume scattering dominates both co- and cross-pol. But in the case of co-pol, there is still a little bit of contribution from rough surface whereas in cross-pol, the contribution is almost entirely from volume scattering.




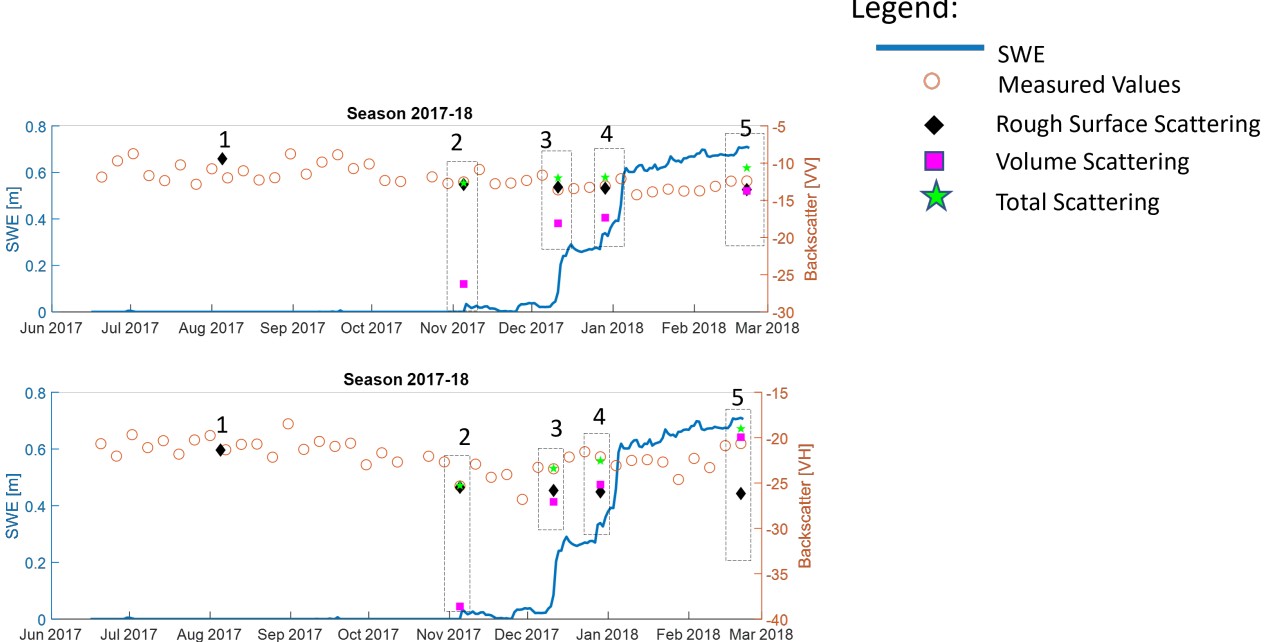

**Figure 9.** Contribution of volume, rough surface and total scattering compared to the measured Sentinel-1 data. At co-pol, the volume and surface scattering compensate each other as the SWE increases resulting in a co-pol backscatter of around -12 dB for the entire snow season. At cross-pol, the volume scattering becomes larger than surface scattering as the SWE increases, resulting in the cross-pol to increase from around -25 dB to -20 dB during the snow season.

Besides SWE, there were no snow microstructure in-situ measurements taken at the site used in this paper. Hence, to determine the parameters of the physical model, best estimates were used based on the time of the season and how realistic the parameter is. For example, at the start of the season, the optical grain size ($11000/<\zeta>$ in mm) is kept at 0.8mm and as the snow increases, the grain size is changed to give the best match to the measured data as the season passes. A similar exercise is done for the other parameters of the bi-continuous DMRT model like density and $b$. To keep the analysis simple, a single layer evolution of the snowpack is used. Soil moisture from SMAP over the same location is used to find the permittivity of the bare soil surface. To find the rms height, the snow free measurements from Sentinel-1 are taken. In this case, the radar is only observing the rough surface (as there is no snow). Since we know the permittivity (from the soil moisture), we can find the optimal rms height for which the simulated backscatter with the NMM3D model matches the observed Sentinel-1 data. By doing this, we find the ground permittivity to be 4.37 + 0.56i (soil moisture of 0.07 $m^3/m^3$) and the rms height to be 1.8 cm. The rms height remains unchanged for the location for the entirety of the season. The soil moisture (soil permittivity) is decreased as the winter onsets. Figure 10 shows the changes in the grain size, density of the snow and the change in soil moisture which gives the match to the measured data showed in Fig 9.



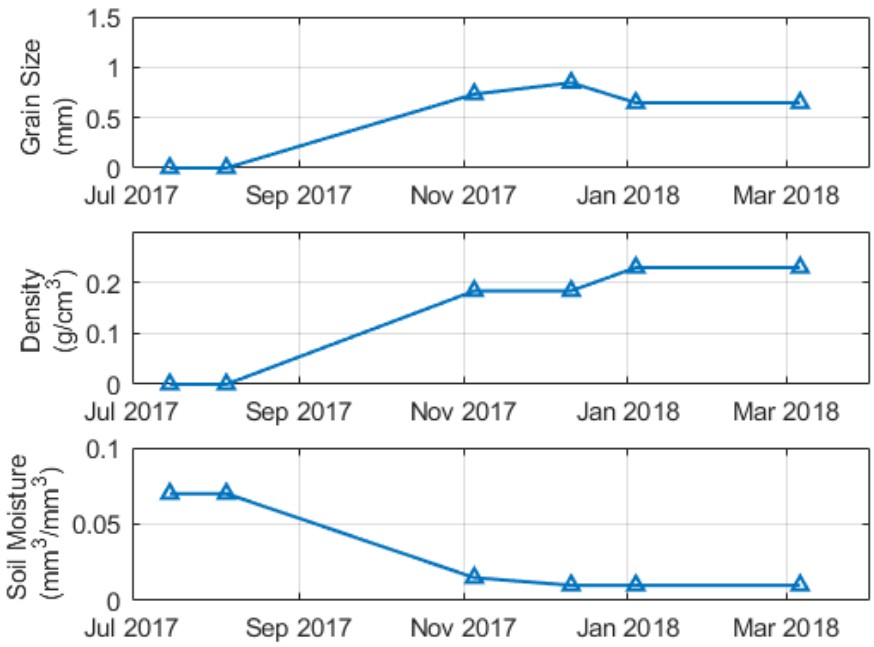

**Figure 10.** Evolution of physical snow and surface parameters throughout the season.

| | | $\sigma_0$ (measured) | Depth (cm) | Attenuation | $\sigma_0^{vol}$ (model) | $\sigma_0^{surf}$ (model) | $\sigma_0^{tot}$ (model) |
|---|---|---|---|---|---|---|---|
| Point 1 | VV | -10.76 | 0 | 0 | 0 | -9.41 | -9.41 |
| | VH | -21.33 | 0 | 0 | 0 | -21.36 | -21.36 |
| Point 2 | VV | -12.52 | 15 | 0.9839 | -26.84 | -12.87 | -12.67 |
| | VH | -25.62 | 15 | 0.9839 | -38.61 | -25.47 | -25.62 |
| Point 3 | VV | -13.59 | 88.8 | 0.909 | -18.08 | -13.21 | -11.99 |
| | VH | -23.43 | 88.8 | 0.909 | -27.07 | -25.81 | -23.43 |
| Point 4 | VV | -13.06 | 178 | 0.8766 | -17.02 | -13.44 | -11.90 |
| | VH | -22.1 | 178 | 0.8766 | -23.88 | -26.04 | -21.54 |
| Point 5 | VV | -12.39 | 384 | 0.8403 | -15.19 | -13.72 | -11.58 |
| | VH | -20.64 | 384 | 0.8403 | -21.36 | -26.32 | -19.01 |

**Table 2.** Measured backscatter compared to the modelled backscatter at each point of Fig 9.

Combining all of this, we can see that initially, rough surface scattering contributes more, and volume scattering contributes
very little, but as the snow starts accumulating, the volume scattering contribution starts increasing and rough surface contribution starts decreasing due to attenuation. In the case of co-pol, the volume scattering compensates for the rough surface



scattering in such a way that the total scattering is relatively unchanged for the snow covered season. Table 2 shows that the co-pol is insensitive to the increase in snow depth and stays constant around -12 dB for the whole snow-covered season. Cross-pol on the other hand is much more sensitive to volume scattering as cross-pol rough surface scattering is much lower compared

to co pol. Hence, cross-pol shows more sensitivity to snow as shown in Table 2. Cross-pol increases from -25 to -20 dB for the snow-covered season. This is why the ratio of cross-pol to co-pol also increases from -13 to -8.5 dB.

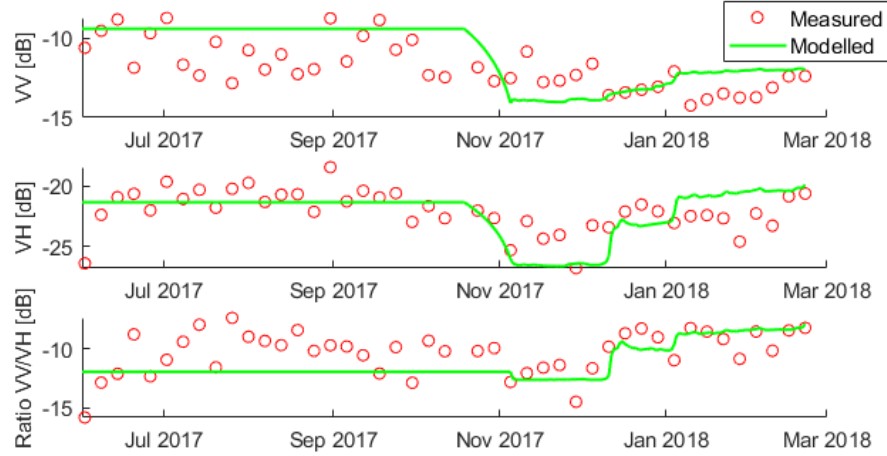

**Figure 11.** Modelled data compared to measured Sentinel-1 data over the 2017-2018 season for Alps location 1 ($44^0$ $40'$ $12''$ N, $6^0$ $25'$ $12''$ E). Co-pol stays around -12 dB. Cross-pol increases from -24 dB to -20 dB. The ratio of cross- to co-pol increases from -13 dB to -8.5 dB.

The analyses of the five points are next extended to the entire seasons by employing appropriate physical and model parameters over the season. We compare the DMRT simulations with Sentinel-1 data taken through the years of 2017 to 2020 for

four different locations as shown in Figures 11, 12, 13 and 14. As stated before, because of the high resolution used (100m) in the analysis, the data shows some sudden jumps and variations. But the model shows the general trend that the measurement data follows. Each location for different years show that the cross-pol and the ratio of cross- to co-pol increases by about 4 to 5 dB whereas the co-pol data remains around -12 dB for the entirety of the snow fall period. In this study, we assume a dry snowpack and a single layer model. In reality, as the snow season goes through, the snow pack will become more complicated.

These complexities may include anisotropy, in-homogeneity or even wet snow. In Figure 11 we can see that the backscatter drops during complete snow period (January 2018) even though SWE is increasing (Fig. 1) which can occur due to increased attenuation from wetness. These complexities are hard to account for without in-situ measurements. As indicated in Section 3 and Lievens et al. (2019), cross-pol signal have small variations for snowpack with depth below 0.5 m. Thus, C-band radar is applicable for the detection of snow with depth above 1 m. For snowpack less than 1 m, other higher bands (X and Ku) are

more suitable since volume scattering is more significant at higher frequencies.





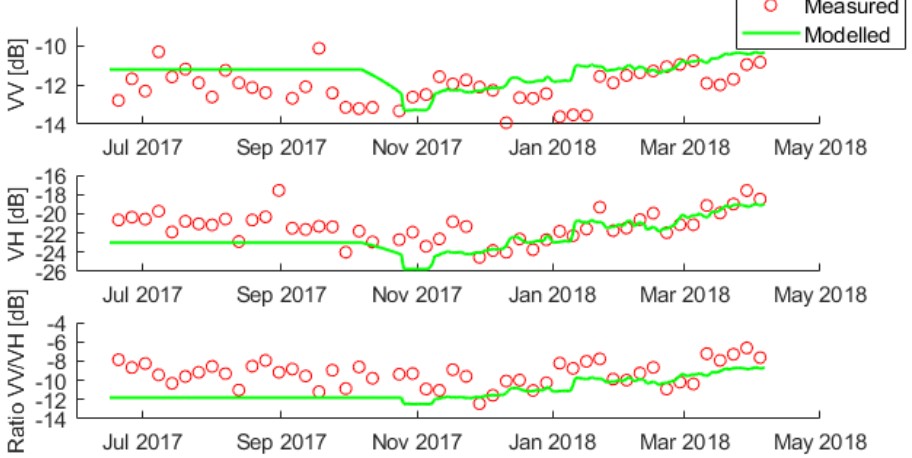

**Figure 12.** Modelled data compared to measured Sentinel-1 data over the 2017-2018 season for Alps location 3 ($44^0$ $40^{'}$ $30.36^{''}$ N, $6^0$ $56^{'}$ $48.84^{''}$ E). Co-pol increases from -12 to -10 dB. Cross-pol increases from -24 dB to -18 dB. The ratio of cross- to co-pol increases from -12 dB to -8 dB.

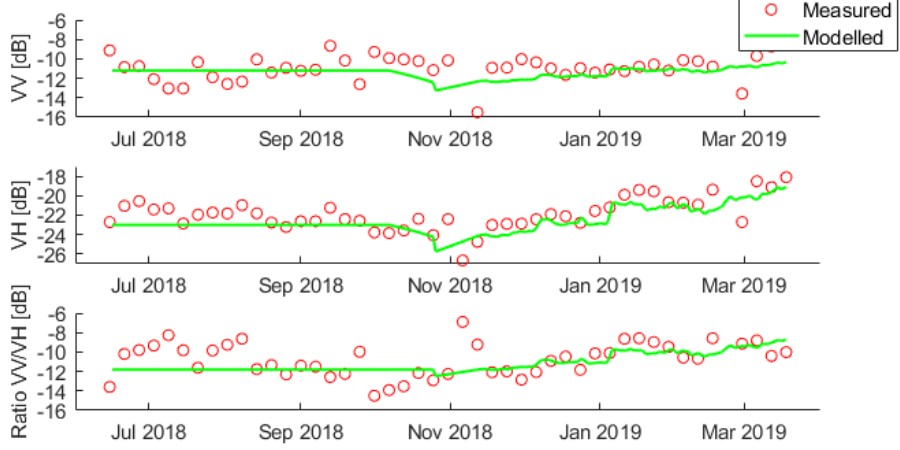

**Figure 13.** Modelled data compared to measured Sentinel-1 data over the 2018-2019 season for Alps location 3 ($44^0$ $40^{'}$ $30.36^{''}$ N, $6^0$ $56^{'}$ $48.84^{''}$ E). Co-pol stays around -11 dB. Cross-pol increases from -24 dB to -19 dB. The ratio of cross- to co-pol increases from -12 dB to -9 dB.

## 5 Conclusions

In this paper, we analyze co-pol and cross-pol data collected by Sentinel-1 over the Alps. We show that combining the bi-continuous DMRT for volume scattering of snow along with NMM3D model for rough surface scattering gives a very good

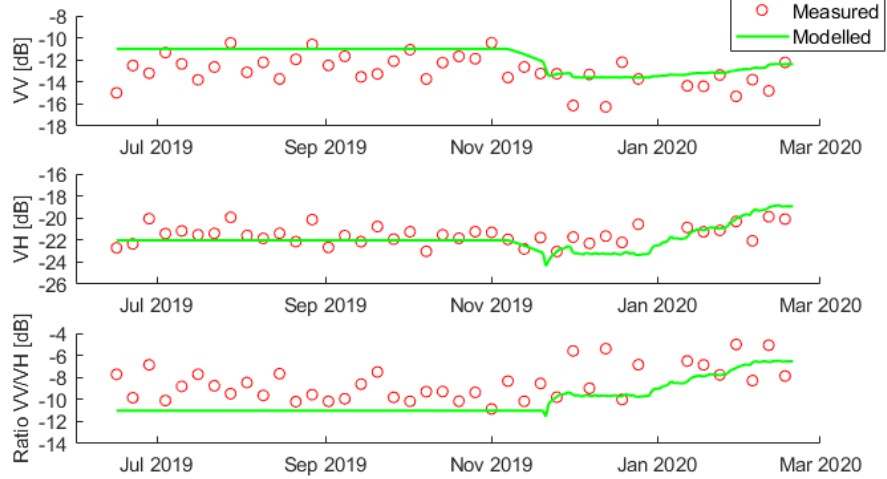

**Figure 14.** Modelled data compared to measured Sentinel-1 data over the 2019-2020 season for Alps location 2 ($45^0$ $17^{'}$ $24.72^{''}$ N, $6^0$ $39^{'}$ $33.12^{''}$ E). Co-pol stays around -12 dB. Cross-pol increases from -24 dB to -19 dB. The ratio of cross- to co-pol increases from -10 dB to -6 dB.

match to the measured data of Sentinel-1. We also show that the increase in volume scattering due to snowfall and decrease

in rough surface scattering due to attenuation happens in such a way that co-pol does not show any sensitivity to increase in snow depth, but the cross-pol shows about 6 dB of dynamic range with respect to SWE. This is also translated into the ratio of cross-pol to co-pol.

The model has been applied to backscatter of terrestrial snow at C-band and validated with radar measurements, but there

is need for further analysis in the future. In this paper, apart from snow depth, rest of the physical and model parameters like $< \zeta >$ and parameter $b$ are assumed, which is not from in situ measurements. Snow anisotropy and in-homogeneities are also not considered. The effects of layering of snow also needs to be studied. In order to have more detailed analysis and advance these models further, for future studies, we need to have radar data with in situ measurements to support the model validation. We also need to study anisotropic snow and the effects of in-homogeneities on cross-pol.

*Code availability.* The code for bi-continuous DMRT and NNM3D is available by contacting the corresponding author (firozb@umich.edu)

*Data availability.* Sentinel-1 data is publicly available from the European Space Agency (ESA)



*Author contributions.* FKB and JFJ did the physical modelling presented in the paper. HL and JFJ processed the backscatter data. FKB and ZH wrote the manuscript. JFJ, LT, HL and EK helped in editing the manuscript

*Competing interests.* The authors declare no competing interests.

*Acknowledgements.* The authors would like to acknowledge National Aeronautics and Space Administration (NASA) THP Program for funding the work presented in this paper.



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
