# Peer review of "Time Series Analysis of C-Band Sentinel-1 SAR Over Mountainous Snow with Physical Models of Volume and Surface Scattering"

_EGUsphere, 2024_

## Referee Comment (RC1)

**General comments**

In this paper the authors present a dense media radiative transfer model to characterize C-band backscatter through a snowpack over a rough soil surface. After the model is demonstrated, modeled backscatter is compared to timeseries of C-band backscatter data from Sentinel-1 at several sites in the European Alps. The general topic of the paper is highly relevant because there are a number of open questions related to the potential for Sentinel-1 measurements of snow properties.

I can identify two main issues with the current manuscript. First, some of the potential conclusions are obscured by a lack of precise language. The terms snow, snow cover, snow packs, and SWE are used somewhat interchangeably throughout the introduction, and in some cases erroneously (see specific comments). Later, much of the text discusses snow in terms of snow depth but figures usually show SWE instead. Specific and precise language is important here so that the community can start to build consensus around the possibilities and limitations of using C-band data to measure snow properties. Issues throughout the manuscript must be addressed before the paper is ready for publication.

The second main issue is that it appears much of the modeling work presented here was already published by Zhu et al. (2023). For example, Figure 5 in this manuscript appears to be an exact replica of Figure 2 in Zhu et al. (2023) and Figure 8 here is a replica of their Figure 11. In my view, much of Section 3 in this manuscript could be replaced by a reference to Zhu et al. instead. In this case the main contribution of the current manuscript would be in Section 4, where the authors compare the modeling results to Sentinel-1 timeseries data at a few locations. Unfortunately this section is underdeveloped and the authors miss an opportunity to address some of the open questions regarding Sentinel-1 snow measurements. For example, Lievens et al. (2019, 2022) already showed that the cross polarization ratio is correlated with snow depth when snow depth is greater than ~1.5-2 m, and more recently Hoppinen et al. (2024) showed in greater detail that this relationship breaks down in snow depths below 1.5 m. This presents several interesting questions as to why Sentinel-1 backscatter might be uncorrelated with snow depth in shallower snow - could the authors use their modeling setup to explore this further? Even if this is not possible (e.g. because of a lack of in situ measurements which the authors acknowledge), the analysis and results here need to be expanded for a more impactful contribution. For example, Figures 11-14 visually compare Sentinel-1 and modeled backscatter but there is no attempt to quantify these results, or to relate them back to the snow conditions at the study locations. Expanding the snow-related analysis would help address important open questions in snow radar remote sensing and make this manuscript more impactful to the broader snow community.

**Specific comments**

Line 31-32: Please provide additional clarification – why is open ground vs vegetation an issue?

Line 35-36: "as there were not enough active snow measurements" – vague

Line 47: "these recent airborne missions" but TerraSAR-X is a satellite platform.

Line 49: Can you briefly provide relevant details of the TSMM platform in the context of the previous discussion in this paragraph? E.g. what sensors/frequencies will be included? Will snow measurements be derived from a backscatter or phase-based approach? Both?

Line 50-69: Please revisit this paragraph and revise with respect to informal and/or redundant language.

Line 52: Lievens et al. (2019) (and the follow-on in 2022) attempt to measure snow depth using Sentinel-1, not SWE as currently stated. This is an important distinction that must be addressed as it brings additional confusion throughout the paper (see comments below).

Line 55: Please provide reference/explanation for 1 meter depth threshold. Is it that higher frequencies are less accurate in deep snow or C-band is less accurate in shallow snow? Perhaps both?

Line 76-77: Please provide a reference if available.

Line 79: Please provide a reference for "sticky sphere model"

Line 85: "As we are dealing with very deep snow" – unclear

Line 87: "can explain the strong cross-pol" – maybe cross-pol response? Or signal?

Line 98-104: Much of this paragraph seems like repeat information from the previous one, including one sentence that is repeated exactly. Perhaps a leftover paragraph from a previous draft? Please revisit and edit/incorporate as appropriate.

Line 106: "the physical model" – please be more specific here. Is this in reference to the bi-continuous DMRT model?

Line 106-115: There seems to be a lot of detailed information here that is not very relevant to the current study. Just a suggestion, but condensing this section may help the reader focus on the details of the numerical methods/models actually used in this study.

Section 2 (Lines 135-156): Imprecise language contributes to additional confusion here. Line 100 states "Cross-polarization observations are used in this paper because they are the basis of snow depth retrieval at C-band (Lievens et al. (2019))" but the figures in this section compare backscatter to SWE. The text in lines 148-150 go on to describe backscatter response to increased snow depth in the figures, even though snow depth is not shown (only SWE).

Lines 138-139: Since Hoppinen et al. (2024) encountered issues reproducing the data processing steps described by Lievens et al. (2022) I strongly suggest a more detailed description of the backscatter data processing. At minimum please include (preferably in an appendix) a description of the software/code used, the original data source, and the specific processing steps. Even better would be releasing the processed backscatter data. I do not see any mention of the processed data in the Data Availability statement.

Line 140-141: Why is the incidence angle important for this study?

Figures 1-3: These are used for essentially illustrative purposes here. I suggest using only one figure here, either from one representative site (put the other two in an appendix if you want) or combine the three sites in one figure somehow.

Line 153-155: Minimizing spatial variability is a reasonable motivation for using Sentinel-1 data at 100 m resolution but how do you square this choice with the following discussion from Lievens et al. (2022, pp 170-171):

> **"The lower accuracy for the [100 m] retrievals can be caused by the larger impacts of (i) radar speckle that is inherent to radar measurements; (ii) geolocation errors and geometric distortions (foreshortening, layover) in the radar images causing location mismatches in $\gamma^0$ time series; and (iii) local heterogeneity in topography (elevation, slope, aspect), land surface properties (land cover, soil moisture, soil temperature, and surface roughness), and snow variables (stratigraphy and microstructure)."**

Several of those factors seem like they would be very important to minimize in this study, especially if these effects are inherent in the validation data but not represented in the model chain. Did you try using the 500 m or 1 km data as in Lievens et al. (2022)?

Figure 4: This looks to be an exact reprint of Figure 1 in Tsang et al. (2022). At a minimum that reference to that paper should be included in the caption.

Line 178: C-band is sensitive to snow in the cross-pol but not in co-pol (instead of snow being sensitive to the backscatter as currently written)

Figure 5: This looks to be an exact reprint of Figure 2 in Zhu et al. (2023). At a minimum that reference to that paper should be included in the caption.

Equation 2: Variable N is not defined.

Section 3.1 and 3.2: A high level of modeling background is assumed in this section. I think there is an opportunity here to modify this text so that people with backgrounds in snow hydrology, different snow remote sensing techniques, etc. can better appreciate the importance of this work. Examples (non-exhaustive) of terms that could use references and/or definitions include discrete dipole approximation (line 201), full-wave simulations (203), scattered far fields (208), realizations averaging (208), eigen-quadrature approach (217) sparse matrix canonical grid (228), near field preconditioning (229), Poggio–Miller–Chang–Harrington–Wu (PMCHW) formulation (234), Rao–Wilton–Glisson (RWG) basis function (238), Galerkin's method (239).

Line 202-203: Remove redundant "Step 1 consists of a generation of volume bi-continuous media."

Line 203: Just use the DDA acronym since it was defined in the previous sentence.

Line 205-206: Move the description of the third step after all discussion of the second step.

Line 238: "The L and K operators are defined in Liao et al. (2016)." It would be helpful to briefly describe them here also.

Equations 5 and 6: In addition to L and K, many terms from this equation are not defined.

Line 242: GMRES acronym not defined.

Line 249-262: The phrasing in this discussion is sometimes too informal and obscures the meaning. E.g. Line 255 "This makes cross-pol more sensitive to the snow depth as volume scattering in cross-pol is dominant."

Line 249: Suggestion: Insert a new section title before this paragraph, e.g. "3.3 Forward Modelling Results"

Line 253: Avoid using phrases like "very high" – 200 cm could be considered shallow or moderate snow depth in some places. In fact you imply this in Line 261. Please check for the phrase throughout the document – there are other instances elsewhere in this paragraph and the captions for Table 1 and Figure 8.

Table 1: Consider removing completely. It seems unnecessary alongside Figure 8.

Figure 8: This looks to be an exact reprint of Figure 11 in Zhu et al. (2023). At a minimum that reference to that paper should be included in the caption.

Section 4: The goal/approach of this section is not immediately clear because there is no introduction or transition from the previous section. After reading below and looking at some figures I can understand that the model was run and compared to Sentinel-1 data, but this should be made explicit as the focus of the section from the beginning.

Lines 264-278: Again the framing of the discussion in terms of snow depth does not match the SWE presented in Figure 9.

Line 271 and 277: "little bit of" → "small"

Line 291-292: "The soil moisture (soil permittivity) is decreased as the winter onsets." Please clarify – was this parameter adjusted in the DMRT model until the results matched the S1 data, or did you use some other empirical/physical model for decreasing soil moisture?

Figures 11-14 would benefit from including the snow depth timeseries as well, to provide context for the timing of the Sentinel-1 measured backscatter.

Figures 12-14 should be combined into one longer timeseries figure for easier comparison between the seasons at the same location.

Line 312-314: "As indicated in Section 3 and Lievens et al. (2019), cross-pol signal have small variations for snowpack with depth below 0.5 m. Thus, C-band radar is applicable for the detection of snow with depth above 1 m." What about snow depths between 0.5-1 m? Also, this sentence has been copied almost verbatim from Zhu et al. (2023, p. 3618).

**Technical corrections**

Throughout document: Check formatting of in-text citations for extra parentheses, e.g. (Kelly et al. (2003)) in Line 25.

Lines 14-15: Suggestion: define the abbreviations "co-pol" and "cross-pol" after the full terms co-polarized and cross-polarized in Line 2. Also check entire document for constancy: "cross-pol" in line 15 vs "cross pol" (no dash) in line 17.

Line 23: Delete "etc"

Line 40: Delete "also"

Line 41: Delete "big" and consider moving this sentence to be the second sentence of the paragraph.

Line 51: Delete "the study of"

Line 79: Acronym QCA is never defined.

Line 129-130: "and the trend that the data are described" – missing a word?

Line 131: "time series data of Sentinel-1 the one season" – unclear

Line 151-152: "was tried to answered" – typo

Line 162: "In 1" → "In (1)"

Line 180: "which in shown by." – unclear/typo

Line 199: Should be Figure 6.

Line 201: "discreet" – typo

Line 207: "discreetized" – typo

Line 219: "raio" – typo

Line 261: constantly → consistently, change 2 m to 200 cm to match the previous presentation

**References**

Hoppinen, Z., Palomaki, R. T., Brencher, G., Dunmire, D., Gagliano, E., Marziliano, A., Tarricone, J., and Marshall, H.-P.: Evaluating Snow Depth Retrievals from Sentinel-1 Volume Scattering over NASA SnowEx Sites, EGUsphere [preprint], 1–35, https://doi.org/10.5194/egusphere-2024-1018, 2024.

Lievens, H., Demuzere, M., Marshall, H.-P., Reichle, R. H., Brucker, L., Brangers, I., de Rosnay, P., Dumont, M., Girotto, M., Immerzeel, W. W., Jonas, T., Kim, E. J., Koch, I., Marty, C., Saloranta, T., Schöber, J., and De Lannoy, G. J. M.: Snow depth variability in the Northern Hemisphere mountains observed from space, Nature Communications, 10, 1–12, https://doi.org/10.1038/s41467-019-12566-y, 2019.

Tsang, L., Durand, M., Derksen, C., Barros, A. P., Kang, D.-H., Lievens, H., Marshall, H.-P., Zhu, J., Johnson, J., King, J., Lemmetyinen, J., Sandells, M., Rutter, N., Siqueira, P., Nolin, A., Osmanoglu, B., Vuyovich, C., Kim, E., Taylor, D., Merkouriadi, I., Brucker, L., Navari, M., Dumont, M., Kelly, R., Kim, R. S., Liao, T.-H., Borah, F., and Xu, X.: Review article: Global monitoring of snow water equivalent using high-frequency radar remote sensing, The Cryosphere, 16, 3531–3573, https://doi.org/10.5194/tc-16-3531-2022, 2022.

Zhu, J., Tsang, L., and Xu, X.: Modeling of Scattering by Dense Random Media Consisting of Particle Clusters With DMRT Bicontinuous, IEEE Transactions on Antennas and Propagation, 71, 3611–3619, https://doi.org/10.1109/TAP.2023.3240562, 2023.

---

## Referee Comment (RC2)

**Comments on: Time Series Analysis of C-Band Sentinel-1 SAR Over Mountainous Snow with Physical Models of Volume and Surface Scattering**

**By Borah F.K., J.-F. Jans, Z. Huang, L. Tsang, H. Lievens, and E. Kim.**

The authors present results of C-band radar backscatter simulations for snow covered ground, applying a bi-continuous dense medium radiative transfer model for computing scattering in the snow volume and the Numerical Solution of Maxwell's Equation in 3-D (NMM3D) method for ground surface scattering. A main motivation for the study is the proof of the hypothesis that in case of deep snow the C-band cross-polarized backscatter contributions of the snow volume exceed the cross-polarized backscatter signal of the ground surface. For tuning of model input data and validation C-band VV- and VH-polarized Sentinel-1 backscatter time series data of three high elevation sites in the French Alps are used. The topic of the paper, C-band radar wave interaction with seasonal snow, is of great relevance for advancing methods for deducing physical snow properties from satellite-borne radar data. However, the manuscript provides a rather narrow view on this topic, lacking actual observations of physical properties of the snow and ground media for model input and validation, as well as lacking specific information on properties of the Sentinel-1 data used at the different locations.

*Main issues:*

The model input data for describing the interaction mechanism of the radar signal with snow and ground are purely hypothetical, not based on observations of physical properties, morphology and temporal evolution of the snowpack, nor on physical properties and state of the ground in the snow-free and snow-covered cases. This kind of information is essential for testing and validating models on radar wave interaction with natural media. The authors selected sites for model testing and validation where such information is not available (except point data on snow depth and SWE), rather than selecting sites at which also other physical snow properties are measured. Suitable data would be available from various snow and avalanche research institutions in the European Alps that regularly collect and publish data on snow stratigraphy, profiles of density, microstructure, grain size and type, hardness, etc. at several sites and on several dates during winter. This would be a useful basis for assessing the response of Sentinel-1 backscatter signals and model performance in respect to specific snow properties. An excellent source of relevant snow and radar data (including Sentinel-1 time series and tower-based C-band radar measurements) is also the multi-year NASA SnowEx program. Authors of this manuscript contributed to this program.

The magnitudes and temporal evolution of the parameters used for model input (Fig. 10) are not in accordance with typical properties of snow and ground in high Alpine terrain of the European Alps. For example, the average size of the scattering elements of the total snowpack decreases during the main accumulation period as long as the snowpack is dry (Calonne et al., 2020). Of particular concern are also the assumptions regarding the soil moisture content. First, it should be mentioned that the selected sites, located in the Alpine tundra zone, are almost completely covered by Alpine grassland and only to a small part by bare surfaces. The soil moisture content in Fig. 10 is largely underestimated. For example, measurements at a Swiss station in 2450 m elevation throughout three years show that the 10 cm moisture content of soil never dropped below 20 vol % (Pellet and Hauck, 2017). In the European Alps major rainfall events up to high altitudes happen also during autumn, and transient melt events are common after the first snowfall. Throughout winter the soil at the snow ground /interface in the Alpine tundra zone of the European Alps contains liquid water because the ground heat creates conditions close to melting and the soil temperature is bounded at 0° C (e.g. Wever et al., 2014; 2015). The lack of a common temporal trend of sigma0 VV during the pre-snowfall and dry snow periods (Figs. 1, 2, 3) is in line with these observations. Furthermore, differences snow and soil properties between individual sites and years may also play a role.

Specification regarding the satellite track and local incidence angle of the Sentinel-1 data at the 3 sites used for input data tuning, validation and shown in the figures is missing. Most areas of the European

Alps are covered by Sentinel-1 IW mode data of four different satellite tracks, which means a single site is viewed within every 12-day period under four different incidence angles and two different aspect angles. In the paper it is not mentioned if the Sentinel-1 data shown in the figures and used in the study are obtained from a single track or are composites of multiple tracks. The incidence angle has a major impact on the partition between volume scattering and surface scattering contributions. Sentinel-1 data of different incidence angles can be used for checking the model performance in this respect.

***Further comments:***

Snow depth and SWE are used alternately in the paper. Please check the proper use of these terms and specify the respective data sources.

Line 47: Please explain to which airborne mission-s this statement refers.

Line 52: Lievens et al (2019) refers to snow depth.

Line 135-136: Please explain why these three sites have been selected, rather than sites with more comprehensive in situ snow observations that include information snow morphology, structure, stratification. Please provide specifications for the snow depth and SWE measurement sensors.

Line 137: The surface cover of these three sites is dominated by vegetation cover (alpine grassland). This can be checked by means of very high resolution satellite imagery.

Line 141ff, Figs 1 ,2, 3: Please provide details on the satellite data shown in these figures and used for validation (single track or merger of several tracks, temporal aggregation method (in case this is applied), local incidence angle, number of looks, radiometric calibration). Also, please explain if SWE in these figures is based on measured data or deduced from snow depth, using density estimates.

Line 144: Please specify the source of information on snow density for relating snow depth and SWE.

Line 161, Fig. 4 and volume scatter simulations: The term on ground-surface/volume interaction is missing.

Line 193 ff: The microstructure representation and derived bi-continuous media formulation should be related to actual observations of microstructure in high Alpine snowpacks and in which way the variations between individual snow layers are taken into account.

Line 225ff, Rough soil surface scattering: The selected soil roughness and dielectric properties do not match the specification of vegetated surfaces, as the case for the three test sites which are covered by Alpine grassland (see main issues).

Line 262 and Table 1: Please specify the incidence angle to which the backscatter values refer.

Figure 8: Results for different incidence angles would be of interest.

Figure 9: Please specify on which Sentinel-1 tracks and sites the sigma0 values are based. Data for season 2017-18 are shown here during which sigma0 shows a different behaviour in the snow accumulation period than in the other years (in particular at point 1).

Line 282ff and Fig. 10: Please explain why sites without in-situ snow microstructure were selected. In order to obtain model input data, a tuning exercise may end up with data on the snowpack and ground that do not match actual properties of snow structure and the ground (see main issues).

Line 286ff: Since September 2015 on SMAP only the microwave radiometer with a footprint size of 39 km x 47 km has been working. Barely the scale for deriving soil moisture and permittivity for sites of 100 m extent in mountainous terrain. Regarding the assumptions for soil moisture content see main comments.

Line 299: The statement "Cross-pol on the other hand is much more sensitive to volume scattering as cross-pol rough surface scattering is much lower compared to co-pol" is not in accordance with the

sigma0 data shown in Figs 1, 2, 3. At sites 1 and 2 sigma0 VH is of similar magnitude during the snow-free and dry snow periods.

Figures 11 to 14: The comparisons of computed and measured sigma0 and the related discussion in the text are not conclusive. Data of different seasons are shown for the individual sites: 2017/18 for point 1, 2018/19 for point 2, 2017/18 and 2018/19 for point 3. Showing data of all years and also related snow depth time series is recommended.

Line 303 ff: For evaluating model performance, quantitative statistical analysis needs to be performed. For comprehensive assessment it is also necessary to show comparisons with backscatter data that are not used for tuning the model input parameters.

***References:***

Calonne, N., Richter, B., Löwe, H., et al.: The RHOSSA campaign: multi-resolution monitoring of the seasonal evolution of the structure and mechanical stability of an alpine snowpack, The Cryosphere, 14, 1829–1848, 2020.

Pellet, C. and Hauck, C.: Monitoring soil moisture from middle to high elevation in Switzerland: set-up and first results from the SOMOMOUNT network, Hydrol. Earth Syst. Sci., 21, 3199–3220, 2017.

Wever, N., Fierz, C., Mitterer, C., Hirashima, H., and Lehning, M.: Solving Richards Equation for snow improves snowpack meltwater runoff estimations in detailed multi-layer snowpack model, The Cryosphere, 8, 257–274, 2014.

Wever, N., Schmid. L., Heilig, A., Eisen, O., Fierz, C., and Lehning, M.: Verification of the multi-layer SNOWPACK model with different water transport schemes, The Cryosphere, 9, 2271–2293, 2015.

---

## Author Comment (AC1)

Response to reviewer's comments-

The responses are in blue text.

The authors present results of C-band radar backscatter simulations for snow covered ground, applying a bi-continuous dense medium radiative transfer model for computing scattering in the snow volume and the Numerical Solution of Maxwell's Equation in 3-D (NMM3D) method for ground surface scattering. A main motivation for the study is the proof of the hypothesis that in case of deep snow the C-band cross-polarized backscatter contributions of the snow volume exceed the cross-polarized backscatter signal of the ground surface. For tuning of model input data and validation C-band VV- and VH-polarized Sentinel-1 backscatter time series data of three high elevation sites in the French Alps are used. The topic of the paper, C-band radar wave interaction with seasonal snow, is of great relevance for advancing methods for deducing physical snow properties from satellite-borne radar data. However, the manuscript provides a rather narrow view on this topic, lacking actual observations of physical properties of the snow and ground media for model input and validation, as well as lacking specific information on properties of the Sentinel-1 data used at the different locations.

***Main issues:***
*The model input data for describing the interaction mechanism of the radar signal with snow and ground are purely hypothetical, not based on observations of physical properties, morphology and temporal evolution of the snowpack, nor on physical properties and state of the ground in the snow-free and snow-covered cases. This kind of information is essential for testing and validating models on radar wave interaction with natural media. The authors selected sites for model testing and validation where such information is not available (except point data on snow depth and SWE), rather than selecting sites at which also other physical snow properties are measured. Suitable data would be available from various snow and avalanche research institutions in the European Alps that regularly collect and publish data on snow stratigraphy, profiles of density, microstructure, grain size and type, hardness, etc. at several sites and on several dates during winter. This would be a useful basis for assessing the response of Sentinel-1 backscatter signals and model performance in respect to specific snow properties. An excellent source of relevant snow and radar data (including Sentinel-1 time series and tower-based C-band radar measurements) is also the multi-year NASA SnowEx program. Authors of this manuscript contributed to this program.*

Thank you for pointing this out. The reason why these particular sites were chosen because our co-authors from the Belgium will do a complete SWE retrieval study of the area of the chosen sites and a few other. That study will use the physical models described in this paper to solve the inverse problem of retrieving SWE. As such, this paper is focused on evaluating the forward model at these Sentinel-1 sites and to physically explain the reason why Sentinel-1 data shows increased sensitivity to cross-polarization backscattering when SWE increases whereas co-polarized backscatter is relatively unchanged. This has been reported in many articles like Lievens et al 2019, 2022 and Hoppinen et al 2024. But they do not give a physical reasoning of this increase. Zhu et al. 2023 showed the physical modelling but did not test it with time series Sentinel-1 data. This paper explains the physical modelling with Sentinel-1 data on how the combination of volume and surface scattering results in the correlation of cross-polarization ratio to SWE. This paper uses SWE but the physical reasoning remains the same. This paper takes the hypothesis of Zhu et al and applies it successfully to the Sentinel-1 data. The goal was to explain the measured Sentinel-1 data with physical modelling of volume and surface scattering. The authors believed that it was more time efficient to use the same locations that will be used in all the studies and hence kept these sites. We realize

that it meant using sites that lack detail in-situ measurements. The work shows that even in lack of detailed information about snow properties information, the model can show the trend of Sentinel-1 data which is important as satellite mission (current or future) may not have access to these detailed in-situ measurements. We corrected the issues with soil moisture in the comments below. Regarding inverting the model for retrieving SWE with many unknown snow parameters, we do not use the complete bi-continuous DMRT as is. We are developing a parameterized single layer bi-continuous DRMT model in which the different parameters of snow are condensed into just two. This results in a very simple model which lets us invert for the co-polarized and cross-polarized backscattered signal to SWE without detailed knowledge of snow parameters or prior information. The details can be found in Borah et al 2023. That work is for two frequencies of X and Ku band, we are modifying it to use for single frequency of C band with two polarization measurements. The work started sometime after this paper and could not be put into this paper. This paper focused on the forward model and physical explanation of the Sentinel-1 data. Quantitative analysis of these results and relating them back to the snow conditions for successful retrieval of SWE is being worked on by the co-authors from the Belgium group and those results will be explained in detail in future work. This explanation of site choice has also been added to the Section-2. Hope this make the authors choice clear to the reviewer and find it satisfactory.

*The magnitudes and temporal evolution of the parameters used for model input (Fig. 10) are not in accordance with typical properties of snow and ground in high Alpine terrain of the European Alps. For example, the average size of the scattering elements of the total snowpack decreases during the main accumulation period as long as the snowpack is dry (Calonne et al., 2020). Of particular concern are also the assumptions regarding the soil moisture content. First, it should be mentioned that the selected sites, located in the Alpine tundra zone, are almost completely covered by Alpine grassland and only to a small part by bare surfaces. The soil moisture content in Fig. 10 is largely underestimated. For example, measurements at a Swiss station in 2450 m elevation throughout three years show that the 10 cm moisture content of soil never dropped below 20 vol % (Pellet and Hauck, 2017). In the European Alps major rainfall events up to high altitudes happen also during autumn, and transient melt events are common after the first snowfall. Throughout winter the soil at the snow ground /interface in the Alpine tundra zone of the European Alps contains liquid water because the ground heat creates conditions close to melting and the soil temperature is bounded at 0° C (e.g. Wever et al., 2014; 2015). The lack of a common temporal trend of sigma0 VV during the pre-snowfall and dry snow periods (Figs. 1, 2, 3) is in line with these observations. Furthermore, differences snow and soil properties between individual sites and years may also play a role.*

The authors would like to thank the reviewer for pointing this information out. First, we would like to point that the sites that we have chosen for study are mostly bare surface with little Alpine grassland verified under satellite image. We have shown two locations here.

[Figure]

We corrected the soil moisture in accordance with Pellet and Hauck, 2017 which shows that the soil moisture is between 40 vol% to 20 vol%. We changed the soil moisture for the analysis of Figure 9. Now the soil moisture for snow free time starts at 35 vol%. Then the soil moisture goes down to 30 vol% and winter sets in. This results in the reduction of backscattered signal at co-polarization and cross-polarization as the data shows in Figure 1 and the model in Figure 9. Since there is little snow fall at this point the backscattered signal is from the surface scattering. A reduction in surface scattering component means reduced soil moisture. As the snow starts accumulating, as stated by the reviewer, the moisture is contained in below the snow pack, the soil moisture is kept at 30 vol% for the remaining of the dry winter period. Please note that, changes to the soil moisture meant adjusting the rms height of the bare soil surface. The revised rms height is now 0.8 cm. This is in accordance with the measured Sentinel-1 data. The updated parameter evaluation graph is shown below-

[Figure]

This parameter evaluation is only for the winter season of 2017-2018 for location 1 as shown in Figure 9. Different years and different loactions have different parameter evaluation conducted in a similar manner as this one but in accordance with the measurements of those years and locations. This has also been made clear in the manuscript in the paragrph between lines 281 till 293. Figure 9 and 10 have been updated in the manuscript but Figure 9 remains very same as the absolute backscatter from the rough surface scattering remains the same.

*Specification regarding the satellite track and local incidence angle of the Sentinel-1 data at the 3 sites used for input data tuning, validation and shown in the figures is missing. Most areas of the European Alps are covered by Sentinel-1 IW mode data of four different satellite tracks, which means a single site is viewed within every 12-day period under four different incidence angles and two different aspect angles. In the paper it is not mentioned if the Sentinel-1 data shown in the figures and used in the study are obtained from a single track or are composites of multiple tracks. The incidence angle has a major impact on the partition between volume scattering and surface scattering contributions. Sentinel-1 data of different incidence angles can be used for checking the model performance in this respect.*

Thank you for pointing this out. It has now been added in the Section-2 of the manuscript that: this study uses only one track of Sentinel-1 data, track 161. This keeps the incidence same. The reason for using this particular track is that the incidence angle is higher than 30 degrees most of the time. This incidence angle and higher values are more useful as the volume and surface scattering can be distinguished in the total scattering signal. This is because the rough surface scattering is specular scattering and decreases considerably as the incidence angle increases but the volume scattering is diffused scattering and remains relatively same for a wide range of incidence angle (Tsang et al 2001). At lower incidence angles, below 20 degrees, the volume and surface scattering signature may look the indistinguishable but at angles of 30 degrees and higher, the surface scattering drops down and make the two signal separate. The variation of incidence angle within this one track is 15 degrees and the model performs well.

***Further comments:***
*Snow depth and SWE are used alternately in the paper. Please check the proper use of these terms and specify the respective data sources.*

Thank you for pointing this out. All the terms have been corrected to SWE as the study is based on SWE. Snow pack is used when describing the snow conditions in the model used.

*Line 47: Please explain to which airborne mission-s this statement refers.*

It has now been added to line 47: "Although these recent airborne missions like the SnowEx campaigns and the AlpSAR campaigns…"

*Line 52: Lievens et al (2019) refers to snow depth.*

Thanks for pointing this out. It has been corrected.

*Line 135-136: Please explain why these three sites have been selected, rather than sites with more comprehensive in situ snow observations that include information snow morphology, structure, stratification. Please provide specifications for the snow depth and SWE measurement sensors.*

Please see the main comment for the reasons for site selection. It has been added in Section-2 that: "The SWE data used in the paper is based on modelled SWE. The data is coming from the SnowClim v1.0 process-based model of Lute et al. (2022) adapted with downscaled Multi-Source Weather (MSWX) and Multi-Source Weighted-Ensemble Precipitation, version 2.8 (MSWEP V2.8) data (Beernaert et al. 2023)." The SWE data has also been added to the data availability section as well.

*Line 137: The surface cover of these three sites is dominated by vegetation cover (alpine grassland). This can be checked by means of very high resolution satellite imagery.*

Please see the main comment on this.

*Line 141ff, Figs 1 ,2, 3: Please provide details on the satellite data shown in these figures and used for validation (single track or merger of several tracks, temporal aggregation method (in case this is applied), local incidence angle, number of looks, radiometric calibration). Also, please explain if SWE in these figures is based on measured data or deduced from snow depth, using density estimates.*

It has been added in Section-2: "This study uses only one track of Sentinel-1 data, track 161. This keeps the incidence same. The reason for using this particular track is that the incidence angle is higher than 30 degrees most of the time. This incidence angle and higher values are more useful as the volume and surface scattering can be distinguished in the total scattering signal. This is because the rough surface scattering is specular scattering and decreases considerably as the incidence angle increases but the volume scattering is diffused scattering and remains relatively same for a wide range of incidence angle (Tsang vol 2). At lower incidence angles, below 20 degrees, the volume and surface scattering signature may look the indistinguishable but at angles of 30 degrees and higher, the surface scattering drops down and make the two signal separate. The variation of incidence angle within this one track is 15 degrees and the model performs well. The SWE data used in the paper is based on modelled SWE. The data is coming from the SnowClim v1.0 process-based model of Lute et al. (2022) adapted with downscaled Multi-Source Weather (MSWX) and Multi-Source Weighted-Ensemble Precipitation, version 2.8 (MSWEP V2.8) data (Beernaert et al. 2023)."

*Line 144: Please specify the source of information on snow density for relating snow depth and SWE.*

In line 144 has been changed to: "The SWE in these regions consistently cross more than 0.6 m (snow depth more than 2 m, based on volume fraction of 30% ice)."

This is done only for illustrative purposes to show the reader how deep the snow is in both SWE and snow depth. 30% volume fraction of ice is a large case scenario.

*Line 161, Fig. 4 and volume scatter simulations: The term on ground-surface/volume interaction is missing.*

It is now added in line 166: "The ground surface interaction is not included in Figure 4 as equation 1 is based on first order scattering between the snow volume and ground surface. Ulaby 2015 shows that the majority of scattering comes from first order term."

*Line 193 ff: The microstructure representation and derived bi-continuous media formulation should be related to actual observations of microstructure in high Alpine snowpacks and in which way the variations between individual snow layers are taken into account.*

As mentioned in lines 285-286, a single layer snow pack model is used in this work. Which means we need to use effective average parameters which will give same backscattered signal as that of the multilayer model.

*Line 225ff, Rough soil surface scattering: The selected soil roughness and dielectric properties do not match the specification of vegetated surfaces, as the case for the three test sites which are covered by Alpine grassland (see main issues).*

Please see the main comments.

*Line 262 and Table 1: Please specify the incidence angle to which the backscatter values refer.*

It is now added: "The incidence angle used of the simulations is 40 degrees."

*Figure 8: Results for different incidence angles would be of interest.*

The results for different incidence angle can be found in Zhu et al 2023 and 2024. It has been added in the manuscript.

*Figure 9: Please specify on which Sentinel-1 tracks and sites the sigma0 values are based. Data for season 2017-18 are shown here during which sigma0 shows a different behaviour in the snow accumulation period than in the other years (in particular at point 1).*

The track has been detailed in Section-2. The data is exactly the same between Figure 1 and Figure 9. It looks stretched out because of the scale. The data in Figure 9 ends on 21 March as indicated in line 266 (dry snow).

[Figure]

*Line 282ff and Fig. 10: Please explain why sites without in-situ snow microstructure were selected. In order to obtain model input data, a tuning exercise may end up with data on the snowpack and ground that do not match actual properties of snow structure and the ground (see main issues).*

Please see the main comments.

*Line 286ff: Since September 2015 on SMAP only the microwave radiometer with a footprint size of 39 km x 47 km has been working. Barely the scale for deriving soil moisture and permittivity for sites of 100 m extent in mountainous terrain. Regarding the assumptions for soil moisture content see main comments.*
As commented by the reviewer this has been removed and the explanation in the main comment regarding soil moisture has been added to the manuscript.

*Line 299: The statement "Cross-pol on the other hand is much more sensitive to volume scattering as cross-pol rough surface scattering is much lower compared to co-pol" is not in accordance with the sigma0 data shown in Figs 1, 2, 3. At sites 1 and 2 sigma0 VH is of similar magnitude during the snow-free and dry snow periods.*

Line 299 has been made clearer: "Cross-polarization is more sensitive to volume scattering as cross-polarization rough surface scattering is much lower than the co-polarized rough surface scattering."

The comparison is between the different polarizations of the volume and surface scattering component at the same time. Table 1 shows that for the dry snow period, the cross-polarized backscatter increases from -25dB to -19dB whereas co-polarization backscatter is about -12dB. Snow free observations are useful to quantify the parameters for the rough surface scattering part as stated in lines 287-290. But are not used to study the properties of the snow. Even though sigma0 VH may show similar magnitude between snow free and dry snow period, we are interested in the increase of the VH backscattered signal in the dry snow period, which can also be seen in Figures 1, 2 and 3.

*Figures 11 to 14: The comparisons of computed and measured sigma0 and the related discussion in the text are not conclusive. Data of different seasons are shown for the individual sites: 2017/18 for point 1, 2018/19 for point 2, 2017/18 and 2018/19 for point 3. Showing data of all years and also related snow depth time series is recommended.*

Because the analysis was done for the dry snow period, the authors feel like putting the years in one single graph will have gaps when the snow becomes wet and till the snow free analysis for the next year starts and not like a continuous graph of measured data in Figures 1, 2 and 3. This may take away from the model performance emphasized in one year which shows how the model behaves with the Sentinel-1 data and explains the data. That is why each seasonal year was done separately. The four figures of Figure 11 to 14 also show that the model can be applied to the remaining years as well.

The authors added the SWE data to the Figures 11-14 so that the analysis is clearer. One figure is showed here, remaining have been updated in the manuscript.

[Figure]

*Line 303 ff: For evaluating model performance, quantitative statistical analysis needs to be performed. For comprehensive assessment it is also necessary to show comparisons with backscatter data that are not used for tuning the model input parameters.*

The reasons of not including quantitative analysis of the model are because of 100m resolution as indicated in the manuscript is so spatial variability is minimized. Another one is that using 100m resolution means using 100m land area for rough surface scattering simulations. This is possible with the current NMM3D model described. But increasing the area to 500m significantly increases the complexity of the problem from an electromagnetics point of view. It is true to remove the aspects speckle, elevation and

slope a larger area is recommended and the authors are working on a new rough surface model that will be able to handle larger areas with slope and elevation change by using fractal surfaces. Future analysis will include that model. The current rough surface model assumes a flat surface. This is one of the main reason this paper focused more on physically explaining the Sentinel-1 data with the best possible models available at the time of writing this manuscript and this does not include quantitative analysis of measured and modelled data at the end. The authors believe that with fluctuations due to 100m resolution will not result in accurate quantitative results. The general trend of the data is very much visible which was the important part of the work. This is also made clear in the Section 4 and Conclusions as well. Point by point quantitative analysis of these results and relating them back to the snow conditions for successful retrieval of SWE is being worked on by the co-authors from the Belgium group and those results will be explained in detail in future work using these improved models.

**References:**

Beernaert E., Jans J.-F., Beck H., Lute A.C., Verhoest N., Lievens H. Data Assimilation of Satellite Retrieved Snow Depth (SD) and Snow Water Equivalent (SWE) into a Snow Model, American Geophysical Union, San Fransisco, California (2023), URL https://agu.confex.com/agu/fm23/meetingapp.cgi/Paper/1334422

Borah, F. K., Tsang, L., and Kim, E. J.: SWE Retrieval Algorithms Based on the Parameterized Bi-continuous DMRT Model without Priors on Grain Size or Scattering Albedo, Progress In Electromagnetics Research, 178, 129–147, https://doi.org/10.2528/PIER23071101, 2023.

Fawwaz Ulaby, David Long; *Microwave Radar and Radiometric Remote Sensing* , Artech, 2015.

Hoppinen, Z., Palomaki, R. T., Brencher, G., Dunmire, D., Gagliano, E., Marziliano, A., Tarricone, J., and Marshall, H.-P.: Evaluating snow depth retrievals from Sentinel-1 volume scattering over NASA SnowEx sites, The Cryosphere, 18, 5407–5430, https://doi.org/10.5194/tc-18-5407-2024, 2024.

Lute A.C., Abatzoglou J., Link T., SnowClim v1.0: high-resolution snow model and data for the western United States, Geosci. Model Dev., 15 (13) (2022), pp. 5045-5071, 10.5194/gmd-15-5045-2022

Pellet, C. and Hauck, C.: Monitoring soil moisture from middle to high elevation in Switzerland: set-up and first results from the SOMOMOUNT network, Hydrol. Earth Syst. Sci., 21, 3199–3220, 2017.

Tsang, L., Kong, J. A., Ding, K.-H., and Ao, C. O.: Scattering of Electromagnetic Waves: Numerical Simulations, John Wiley Sons, Ltd, ISBN 9780471224303, https://doi.org/https://doi.org/10.1002/0471224308.ch6, 2001.

---

## Author Comment (AC2)

Response to reviewer's comments-

The responses are in blue text.

*I can identify two main issues with the current manuscript. First, some of the potential conclusions are obscured by a lack of precise language. The terms snow, snow cover, snow packs, and SWE are used somewhat interchangeably throughout the introduction, and in some cases erroneously (see specific comments). Later, much of the text discusses snow in terms of snow depth but figures usually show SWE instead. Specific and precise language is important here so that the community can start to build consensus around the possibilities and limitations of using C-band data to measure snow properties. Issues throughout the manuscript must be addressed before the paper is ready for publication.*

Thank you for pointing this out. All the terms have been corrected to SWE as the study is based on SWE. Snow pack is used when describing the snow conditions in the model used.

*The second main issue is that it appears much of the modeling work presented here was already published by Zhu et al. (2023). For example, Figure 5 in this manuscript appears to be an exact replica of Figure 2 in Zhu et al. (2023) and Figure 8 here is a replica of their Figure 11. In my view, much of Section 3 in this manuscript could be replaced by a reference to Zhu et al. instead.*

The importance of the section for volume scattering is to show the hypothesis that the aggregation of ice crystals when the snow pack is beyond 1.5m deep causes the co-polarization and cross-polarization backscatter to increase and then compare the volume scattering to the rough surface scattering. The physical modelling of dense aggregate with bi-continuous DMRT model was described in detail in Zhu et al. 2023 but was also put here so that the readers understand the model's working when it is applied to the Sentinel-1 data. The main objective of this paper is to use the physical model developed in Zhu et al and apply it to Sentinel-1 data and explain the physical reason of the cross-polarization ratio to snow depth. Authors agree with the reviewer and details of the section has been removed and changed. Please see the revised Section 3 at the end of the response document.

*In this case the main contribution of the current manuscript would be in Section 4, where the authors compare the modeling results to Sentinel-1 timeseries data at a few locations. Unfortunately this section is underdeveloped and the authors miss an opportunity to address some of the open questions regarding Sentinel-1 snow measurements. For example, Lievens et al. (2019, 2022) already showed that the cross polarization ratio is correlated with snow depth when snow depth is greater than ~1.5-2 m, and more recently Hoppinen et al. (2024) showed in greater detail that this relationship breaks down in snow depths below 1.5 m. This presents several interesting questions as to why Sentinel-1 backscatter might be uncorrelated with snow depth in shallower snow - could the authors use their modeling setup to explore this further? Even if this is not possible (e.g. because of a lack of in situ measurements which the authors acknowledge), the analysis and results here need to be expanded for a more impactful contribution. For example, Figures 11-14 visually compare Sentinel-1 and modeled backscatter but there is no attempt to quantify these results, or to relate them back to the snow conditions at the study locations. Expanding the snow-related analysis would help address important open questions in snow radar remote sensing and make this manuscript more impactful to the broader snow community.*

The main objective of this manuscript is to explain the physical reason for the correlation of the cross-polarization ratio with snow depth as reported by Lievens et al. (2019, 2022) and why the relationship breaks down for snow depths less than 1.5m. In the Section 3 from line 250 to 263 describes the combined effect of volume and surface scattering when the snow depth is 0.5m and when the snow depth is 2m (Table 1) and also in Fig 11 of Zhu et al. 2023. This shows that when the snow depth is less than 1.5m, the

surface scattering dominates both the co-polarization and cross-polarization backscatter and since surface scattering does not increase with snow depth (it decreases slightly due to attenuation from the snow pack as shown in Fig 11 of Zhu et al. 2023 and Table 1), the cross-polarization ratio is uncorrelated to snow depth when the snow depth is less than 1.5m. On the other hand, when the snow depth increases beyond 1.5 to 2m, the volume and surface scattering of co-polarization backscatter becomes comparable to one another. The co-polarization volume scattering increases and the surface scattering decreases slightly such that the net result of the total co-polarization backscatter (sum of volume and surface scattering) is uncorrelated to the snow depth. In case of cross-polarization backscatter, the volume scattering become much larger than the surface scattering and beyond 1.5m of snow depth cross-polarization volume scattering dominates. Hence, the total cross-polarization signal becomes correlated to snow depth. As snow depth increases the cross-polarization volume backscatter increases. This results in the ratio of cross-polarization to co-polarization to increase with snow depth. Co-polarization is uncorrelated and cross-polarization increases with snow depth, so, the cross-polarization ratio increases with snow depth. This is reported in Lievens et al 2019, 2022 and Hoppinen et al 2024. Zhu et al. 2023 showed the physical modelling but did not test it extensively with Sentinel-1 data. This paper explains the physical modelling with Sentinel-1 data in Section-4 on how the combination of volume and surface scattering results in the correlation of cross-polarization ratio to SWE. This paper uses SWE but the physical reasoning remains the same. This paper takes the hypothesis of Zhu et al and applies it successfully to the Sentinel-1 data. The goal was to explain the measured Sentinel-1 data with physical modelling of volume and surface scattering. Quantitative analysis of these results and relating them back to the snow conditions for successful retrieval of SWE is being worked on by the co-authors from the Belgium group and those results will be explained in detail in future work.

**Specific comments**

*Line 31-32: Please provide additional clarification – why is open ground vs vegetation an issue?*

It is now added in the manuscript in line 31: "A radiometer will see the total emissions from the scene. With a spatial resolution in the kilometer scale, the total emission will be an average value of all the components in the kilometer scale resolution such as open ground, snow covered vegetation and even water bodies etc. It is not possible to separate all these different components within a kilometer pixel and an accurate study of snow is very difficult in this case."

*Line 35-36: "as there were not enough active snow measurements" – vague*

It is now added in the manuscript in line 35: "active measurements of snow were largely theoretical, such as Tsang and Kong (1992), Mätzler (1998), as there were not enough detailed ground measurements of snow stratigraphy like density, grain size, SSA along with co-located radar measurements to validate these theories extensively."

*Line 47: "these recent airborne missions" but TerraSAR-X is a satellite platform.*

The term airborne has been removed. Thank you for pointing this out.

*Line 49: Can you briefly provide relevant details of the TSMM platform in the context of the previous discussion in this paragraph? E.g. what sensors/frequencies will be included? Will snow measurements be derived from a backscatter or phase-based approach? Both?*

It is now added in the manuscript in line 49: "TSMM will use dual frequency backscatter measurements of low Ku (13.5 GHz) and high Ku (17.2 GHz) to retrieve SWE across northern hemisphere every 7 days (Derkson et al. 2021)."

*Line 50-69: Please revisit this paragraph and revise with respect to informal and/or redundant language.*

The paragraph is now changed to "An alternative satellite mission to monitor and map snow globally is Sentinel-1, launched by the European Space Agency (ESA) and Copernicus. It has been shown that the study of Sentinel-1 cross-polarization backscatter data can provide important retrieval of snow depth in mountainous snow (Lievens et al. 2019). In this work, we analyze the physics behind the cross-polarization backscatter of C-band using accurate physical forward models which in turn can be used to retrieve SWE. Recent studies also show how Sentinel-1 in conjecture with reanalysis SWE can aslo be used to monitor the snow cover (Lei et al. (2023)). The motivation of this work is that Lievens et al. 2019, 2022 showed the ability of Sentinel-1 C-band radar data to map snow depth in the Northern Hemisphere mountains. It was believed that snow volume scattering at C-band is much smaller than surface scattering of the underlying rough snow/soil interface (Nagler and Rott (2000)). This means that C-band signal should be dominated by the rough snow/soil interface backscatter and should be uncorrelated to the snow depth. The two contributions of radar scattering from snow is shown in Fig. 4. But the data from Lievens et al. 2019, 2022 and Hoppinen et al. 2024 show that C-band cross-polarization backscatter increases with snow depth when the snow depth is beyond 1.5m to 2m. These results have aroused great interests because they indicate the possibility of using cross-polarization backscatter of C-band radar data to monitor global snow depth in mountainous regions. However, the physical reasons that cause the strong cross-pol are still unclear for the snow community. It may be caused by the anisotropy of ice particles, ice clusters, or in-homogeneities in the snowpack. In this paper, we explain the strong cross-polarization backscatter sensitivity to snow accumulation in mountainous regions by dense aggregates of ice clusters combined with the effect of rough soil surface scattering below the snow pack. For forward modelling, this paper uses the bi-continuous DMRT model with dense aggregates of ice grain clusters to calculate the volume scattering from the snowpack and NMM3D model to simulate the rough surface scattering. The goal of this paper is to explain the physical reason behind the increase of cross-polarization to co-polarization backscatter ratio with increase in SWE. Once an accurate forward model has been established, it can be then used to retrieve SWE in future studies. Please note that the work done by Lievens et al. 2019, 2022 was based on snow depth, but we use SWE as the basis for our analysis so that it can be used for SWE retrievals.

Authors hope this paragraph is much more clear to the reviewer now.

*Line 52: Lievens et al. (2019) (and the follow-on in 2022) attempt to measure snow depth using Sentinel-1, not SWE as currently stated. This is an important distinction that must be addressed as it brings additional confusion throughout the paper (see comments below).*

Thank you for pointing this out. It has been corrected as a response to the previous comment.

*Line 55: Please provide reference/explanation for 1 meter depth threshold. Is it that higher frequencies are less accurate in deep snow or C-band is less accurate in shallow snow?*

*Perhaps both?*

The statement has been removed as it was ambiguous. But the reason is that at deep snow depths, higher frequencies start to saturate and give more error in retrievals (Borah et al. 2023). C-band cross-polarization backscatter is sensitive to snow depth beyond 1.5 to 2m and can be used for snow monitoring in these snow depths. The statement was removed because the threshold for saturation of higher frequencies needs more sensitivity analysis and that was not part of this paper. 1m threshold can give wrong idea to readers without proper context and hence the statement was removed.

*Line 76-77: Please provide a reference if available.*

The line is changed to: "As these empirical fits are based on measurements from few select places, they may not be able to characterize the snow microstructure in general globally."

*Line 79: Please provide a reference for "sticky sphere model"*

Reference has been added (Tsang et al. 1992).

*Line 85: "As we are dealing with very deep snow" – unclear*

The line is changed to: "In this work, as we are consistently dealing with SWE of more than 600mm, the ice crystals in the snow pack can very easily form dense ice aggregates as hypothesized in Lievens et al. (2019), King et al. (2015), which are highly irregular in shape and which can explain the strong cross-polarized backscatter signal when the SWE is high."

*Line 87: "can explain the strong cross-pol" – maybe cross-pol response? Or signal?*

The term cross-pol and co-pol have been changed to cross-polarized backscatter and co-polarized backscatter to make the language clearer.

*Line 98-104: Much of this paragraph seems like repeat information from the previous one, including one sentence that is repeated exactly. Perhaps a leftover paragraph from a previous draft? Please revisit and edit/incorporate as appropriate.*

Thank you for pointing this out. The paragraph has been changed.

*Line 106: "the physical model" – please be more specific here. Is this in reference to the bicontinuous DMRT model?*

It is changed to "The rough surface component of the physical model" to make it clear.

*Line 106-115: There seems to be a lot of detailed information here that is not very relevant to the current study. Just a suggestion, but condensing this section may help the reader focus on the details of the numerical methods/models actually used in this study.*

The authors feel like putting down the limitations of past models and why they should not be used is important. Even in Zhu et al. 2023 Oh's model was used and that is the reason the work was extended to the NMM3D used in this work. It is more accurate for cross-polarization backscatter.

*Section 2 (Lines 135-156): Imprecise language contributes to additional confusion here. Line 100 states "Cross-polarization observations are used in this paper because they are the basis of snow depth retrieval at C-band (Lievens et al. (2019))" but the figures in this section compare backscatter to SWE. The text in lines 148-150 go on to describe backscatter response to increased snow depth in the figures, even though snow depth is not shown (only SWE).*

Thank you for pointing this out. While the work in Lievens et al 2019 is based on snow depth, this paper focuses on SWE. Hence line 100 is changed to "Cross-polarization observations are used in this paper because they are the basis of snow depth retrieval at C-band (Lievens et al. 2019). We show in this paper that the cross-polarization observations can also be used for SWE retrievals." In line 148-150 we change everything to SWE as this study is on SWE and the explanation of increase in cross-polarization backscatter to increase in SWE in the Sentinel-1 data.

*Lines 138-139: Since Hoppinen et al. (2024) encountered issues reproducing the data processing steps described by Lievens et al. (2022) I strongly suggest a more detailed description of the backscatter data processing. At minimum please include (preferably in an appendix) a description of the software/code used, the original data source, and the specific processing steps. Even better would be releasing the processed backscatter data. I do not see any mention of the processed data in the Data Availability statement.*

It is now added in line 139: "Data obtained from ASF (Alaska Space Facility) data portal (IW data, GRD observations) that were then processed to their gamma0 format in the ESA SNAP toolbox using the following processing steps: precise orbit file application, GRD border and thermal noise removal, radiometric calibration, multilooking, terrain flattening to backscatter as gamma0, range-Doppler terrain correction and projection to WGS 84 (last step within MATLAB). For both the terrain correction and flattening the GLO-30 DEM was used."

*Line 140-141: Why is the incidence angle important for this study?*

It is now added in the text: "Incidence angle plays an important role in rough surface scattering as rough surface scattering depends strongly depends on incidence angle(Tsang et al 2001). We study only one track of Sentinel-1 data (track 161) and that gives one angle for each site that is presented in this paper. The reason for using this particular track is that the incidence angle is higher than 30 degrees most of the time. This incidence angle and higher values are more useful as the volume and surface scattering can be distinguished in the total scattering signal. This is because the rough surface scattering is specular scattering and decreases considerably as the incidence angle increases but the volume scattering is diffused scattering and remains relatively same for a wide range of incidence angle (Tsang et al 2001). At lower incidence angles, below 20 degrees, the volume and surface scattering signature may look the indistinguishable but at angles of 30 degrees and higher, the surface scattering drops down and make the two signal separate. The variation of incidence angle within this one track is 15 degrees and the model performs well."

*Figures 1-3: These are used for essentially illustrative purposes here. I suggest using only one figure here, either from one representative site (put the other two in an appendix if you*

*want) or combine the three sites in one figure somehow.*

It has been changed to show one figure from the site used in the analysis.

*Line 153-155: Minimizing spatial variability is a reasonable motivation for using Sentinel-1 data at 100 m resolution but how do you square this choice with the following discussion from Lievens et al. (2022, pp 170-171):*
   *"The lower accuracy for the [100 m] retrievals can be caused by the larger impacts of (i) radar speckle that is inherent to radar measurements; (ii) geolocation errors and geometric distortions (foreshortening, layover) in the radar images causing location mismatches in γ0 time series; and (iii) local heterogeneity in topography (elevation, slope, aspect), land surface properties (land cover, soil moisture, soil temperature, and surface roughness), and snow variables (stratigraphy and microstructure)."*
*Several of those factors seem like they would be very important to minimize in this study, especially if these egects are inherent in the validation data but not represented in the model chain. Did you try using the 500 m or 1 km data as in Lievens et al. (2022)?*

The reason for using 100m resolution as indicated in the manuscript is so spatial variability is minimized. Another reason is that using 100m resolution means using 100m land area for rough surface scattering simulations. This is possible with the current NMM3D model described. But increasing the area to 500m significantly increases the complexity of the problem from an electromagnetics point of view. It is true to remove the aspects speckle, elevation and slope a larger area is recommended and the authors are working on a new rough surface model that will be able to handle larger areas with slope and elevation change by using fractal surfaces. Future analysis will include that model. The current rough surface model assumes a flat surface. This is one of the main reason this paper focused more on physically explaining the Sentinel-1 data with the best possible models available at the time of writing this manuscript and this does not include quantitative analysis of measured and modelled data at the end. The authors believe that with fluctuations due to 100m resolution will not result in accurate quantitative results. The general trend of the data is very much visible which was the important part of the work. This is also made clear in the Section 4 and Conclusions as well.

*Figure 4: This looks to be an exact reprint of Figure 1 in Tsang et al. (2022). At a minimum that reference to that paper should be included in the caption.*

It is now added in line 166: "The figure for different contributions of snow and rough surface backscattering scattering is from Tsang et. al. (2022)."

*Line 178: C-band is sensitive to snow in the cross-pol but not in co-pol (instead of snow being sensitive to the backscatter as currently written)*

Thank you for pointing this out. It has been corrected.

*Figure 5: This looks to be an exact reprint of Figure 2 in Zhu et al. (2023). At a minimum that reference to that paper should be included in the caption.*

Section 3 has been changed. Please see the end of the document. The figure has been removed.

*Equation 2: Variable N is not defined.*

The equation is removed, and details can be found in Zhu et al. 2023. Section 3 has been changed. Please see the end of the document.

*Section 3.1 and 3.2: A high level of modeling background is assumed in this section. I think there is an opportunity here to modify this text so that people with backgrounds in snow hydrology, digerent snow remote sensing techniques, etc. can better appreciate the importance of this work. Examples (non-exhaustive) of terms that could use references and/or definitions include discrete dipole approximation (line 201), full-wave simulations (203), scattered far fields (208), realizations averaging (208), eigen-quadrature approach (217) sparse matrix canonical grid (228), near field preconditioning (229), Poggio–Miller–Chang–Harrington–Wu (PMCHW) formulation (234), Rao–Wilton–Glisson (RWG) basis function (238), Galerkin's method (239).*

The Section 3 has been changed so that it is more relatable to the readers of this journal whose backgrounds are in snow hydrology.

*Line 202-203: Remove redundant "Step 1 consists of a generation of volume bi-continuous media."*

Please see the detailed change to the entire Section 3 at the end.

*Line 203: Just use the DDA acronym since it was defined in the previous sentence.*

Please see the detailed change to the entire Section 3 at the end.

*Line 205-206: Move the description of the third step after all discussion of the second step.*

Please see the detailed change to the entire Section 3 at the end.

*Line 238: "The L and K operators are defined in Liao et al. (2016)." It would be helpful to briefly describe them here also.*

Please see the detailed change to the entire Section 3 at the end.

*Equations 5 and 6: In addition to L and K, many terms from this equation are not defined.*

Please see the detailed change to the entire Section 3 at the end.
*Line 242: GMRES acronym not defined.*

Please see the detailed change to the entire Section 3 at the end.

*Line 249-262: The phrasing in this discussion is sometimes too informal and obscures the meaning. E.g. Line 255 "This makes cross-pol more sensitive to the snow depth as volume scattering in cross-pol is dominant."*

Please see the detailed change to the entire Section 3 at the end.

*Line 249: Suggestion: Insert a new section title before this paragraph, e.g. "3.3 Forward Modelling Results"*

Please see the detailed change to the entire Section 3 at the end.

*Line 253: Avoid using phrases like "very high" – 200 cm could be considered shallow or moderate snow depth in some places. In fact you imply this in Line 261. Please check for the phrase throughout the document – there are other instances elsewhere in this paragraph and the captions for Table 1 and Figure 8.*

Thank you for pointing this out. The phrase has been removed.

*Table 1: Consider removing completely. It seems unnecessary alongside Figure 8.*

Please see the detailed change to the entire Section 3 at the end.

*Figure 8: This looks to be an exact reprint of Figure 11 in Zhu et al. (2023). At a minimum that reference to that paper should be included in the caption.*

Please see the detailed change to the entire Section 3 at the end.

*Section 4: The goal/approach of this section is not immediately clear because there is no introduction or transition from the previous section. After reading below and looking at some figures I can understand that the model was run and compared to Sentinel-1 data, but this should be made explicit as the focus of the section from the beginning.*

It has been added to line 264: "In this section we use the volume scattering physical model of bi-continuous DRMT and the rough surface scattering physical model of NMM3D and apply them to the Sentinel-1 data. We show the combined effect of volume and surface scattering to both co-polarization signal and cross-polarization signal and how the two scattering components contribute together to the increase in the cross-polarized backscattered signal when SWE is increasing. The locations of the study area are in Section 2."

*Lines 264-278: Again the framing of the discussion in terms of snow depth does not match the SWE presented in Figure 9.*

It has been changed to SWE.

*Line 271 and 277: "little bit of" à "small"*

It has been changed.

*Line 291-292: "The soil moisture (soil permittivity) is decreased as the winter onsets." Please clarify – was this parameter adjusted in the DMRT model until the results matched the S1 data, or did you use some other empirical/physical model for decreasing soil moisture?*

We corrected the soil moisture in accordance with Pellet and Hauck, 2017 which shows that the soil moisture is between 40 vol% to 20 vol%. We changed the soil moisture for the analysis of Figure 9. Now the soil moisture for snow free time starts at 35 vol%. Then the soil moisture goes down to 30 vol% and winter sets in. This results in the reduction of backscattered signal at co-polarization and cross-polarization as the data shows in Figure 1 and the model in Figure 9. Since there is little snow fall at this point the backscattered signal is from the surface scattering. A reduction in surface scattering component means reduced soil moisture. As the snow starts accumulating, as stated by the reviewer, the moisture is contained in below the snow pack, the soil moisture is kept at 30 vol% for the remaining of the dry winter period. Please note that, changes to the soil moisture meant adjusting the rms height of the bare soil surface. The revised rms height is now 0.8 cm. This is in accordance with the measured Sentinel-1 data.

*Figures 11-14 would benefit from including the snow depth timeseries as well, to provide context for the timing of the Sentinel-1 measured backscatter.*

The authors added the SWE data to the Figures 11-14 so that the analysis is clearer. One figure is showed here, remaining have been updated in the manuscript.

[Figure]

*Figures 12-14 should be combined into one longer timeseries figure for easier comparison between the seasons at the same location.*

Because the analysis was done for the dry snow period, the authors feel like putting the years in one single graph will have gaps when the snow becomes wet and till the snow free analysis for the next year starts and not like a continuous graph of measured data in Figures 1, 2 and 3. This may take away from the model performance emphasized in one year which shows how the model behaves with the Sentinel-1 data and explains the data. That is why each seasonal year was done separately. The four figures of Figure 11 to 14 also show that the model can be applied to the remaining years as well.

*Line 312-314: "As indicated in Section 3 and Lievens et al. (2019), cross-pol signal have small variations for snowpack with depth below 0.5 m. Thus, C-band radar is applicable for the detection of snow with depth above 1 m." What about snow depths between 0.5-1 m? Also, this sentence has been copied almost verbatim from Zhu et al. (2023, p. 3618).*

The statement has been changed to "From Zhu et al 2023 and Lievens et al. 2019, there is little variation in the C band backscattered signal at both co-polarization and cross-polarization at snow depths between 0.5m to 1m due to dominance of rough surface scattering at these depths. Because of this, C band is more feasible beyond 1m of snow depth when the cross-polarization volume backscatter starts increasing compared to the cross-polarized rough surface scattering as indicated in Section 4."

Anything below 1m of snow depth, C band will not be very useful and different techniques should be used like the ones showed in Borah et al 2023 and Durand at al 2024.

*Technical corrections*

Thank you for pointing out the technical corrections. All of the ones mentioned here and some other have been corrected to the authors full extent.

*Throughout document: Check formatting of in-text citations for extra parentheses, e.g.*
*(Kelly et al. (2003)) in Line 25.*
*Lines 14-15: Suggestion: define the abbreviations "co-pol" and "cross-pol" after the full*
*terms co-polarized and cross-polarized in Line 2. Also check entire document for*
*constancy: "cross-pol" in line 15 vs "cross pol" (no dash) in line 17.*
*Line 23: Delete "etc"*
*Line 40: Delete "also"*
*Line 41: Delete "big" and consider moving this sentence to be the second sentence of the*
*paragraph.*
*Line 51: Delete "the study of"*
*Line 79: Acronym QCA is never defined.*
*Line 129-130: "and the trend that the data are described" – missing a word?*
*Line 131: "time series data of Sentinel-1 the one season" – unclear*
*Line 151-152: "was tried to answered" – typo*
*Line 162: "In 1" à "In (1)"*
*Line 180: "which in shown by." – unclear/typo*
*Line 199: Should be Figure 6.*
*Line 201: "discreet" – typo*
*Line 207: "discreetized" – typo*
*Line 219: "raio" – typo*
*Line 261: constantly à consistently, change 2 m to 200 cm to match the previous*
*presentation*

References:

C. Derksen et al., "Development of the Terrestrial Snow Mass Mission," 2021 IEEE International Geoscience and Remote Sensing Symposium IGARSS, Brussels, Belgium, 2021, pp. 614-617, doi: 10.1109/IGARSS47720.2021.9553496.

Tsang, L. and Kong, J.: Scattering of Electromagnetic Waves from a Dense Medium Consisting of Correlated Mie Scatterers with Size Distributions and Applications to Dry Snow, Journal of Electromagnetic Waves and Applications, 6, 265–286, https://doi.org/10.1163/156939392X01156, 1992.

Tsang, L., Kong, J. A., Ding, K.-H., and Ao, C. O.: Scattering of Electromagnetic Waves: Numerical Simulations, John Wiley Sons, Ltd, ISBN 9780471224303, https://doi.org/https://doi.org/10.1002/0471224308.ch6, 2001.

**3 Forward Modelling at C-band of Volume Scattering of Snow and Surface Scattering of Rough Soil Surface**

The Sentinel-1 radar measures the total backscatter of the illuminated area. This means that it is measuring both the volume scattering (the scattering from the snow pack itself) and the rough soil surface scattering. The total scattering contribution from both the snow volume and rough surface can be given by

$$\sigma^{tot} = \sigma^{vol} + \sigma^{rs} e^{-2\tau \sec \theta_t} \tag{1}$$

In eq. 1, $\sigma^{vol}$ is the volume scattering calculated from the bi-continuous DMRT model. The bi-continuous DMRT is solved numerically so that $\sigma^{vol}$ includes volume multiple scattering effects. The quantity $\sigma^{rs}$ is the surface scattering of rough soil surface from calculated from the NMM3D model. The $e^{-2\tau \sec \theta_t}$ term is the attenuation of the wave travelling through the snow pack. It depends on the optical depth $\tau$ of the snow pack, which can be found from the bi-continuous media DMRT model, and the transmitted angle $\theta_t$ can be found from the Snell's law at the air snow interface. This is shown in Fig. 1. The figure for different contributions of snow and rough surface backscattering scattering is from Tsang et al. (2022). The ground surface interaction is not included in Figure 1 as equation 1 is based on first order scattering between the snow volume and ground surface. Ulaby and Stiles (1980) shows that the majority of scattering comes from first order term.

[Figure]

**Figure 1.** Volume and surface scattering interactions as seen by a radar. The volume contribution $\sigma_{vol}$ comes from the near and far field interactions of the densely packed ice clusters and the surface contribution $\sigma_{bg}$ comes from the bare rough surface scattering $\sigma_{rs}$ attenuated by the snow on top.

**3.1 Volume Scattering**

The bi-continuous DMRT model is used to calculate the volume scattering from the snow. In the bi-continuous DMRT model, first a bi-continuous two-phase random medium that consists of ice particles with a relative permittivity of 3.2 and air background is generated to represent the snow pack. The physical characterization of the snow is controlled by two parameters, $<\zeta>$ and $b$ which correspond to the inverse of mean optical grain size and aggregation of the ice crystals, respectively. The smaller the value of the parameter $b$ is, the larger and more denser are the aggregates. Once the bi-continuous media has been generated, the phase matrix of the bi-continuous media is calculated using full wave simulations of Maxwell's equations. The phase matrix is then used in the DMRT equations to predict the backscatter from snow based on inverse mean grain size ($<\zeta>$), aggregation $b$, snow depth, snow density, and the incidence angle of the incoming wave. The bi-continuous media model was proposed in Ding et al. (2010).

25     As shown by the Sentinel-1 data, C-band in the mountainous snow has a higher sensitivity to cross-pol but very little sensitivity to co-pol. This is due to a combination of rough surface scattering and dense aggregation in the snow pack which can occur in the deep mountainous snow which was shown by Lievens et al. (2019) and Zhu et al. (2023). Rough surface scattering will be discussed in detail in the next section. Zhu et al. (2023) provided the model for the dense aggregation in the snow pack called the tuned bi-continuous DMRT model. That model is similar to the ones developed by Ding et al. (2010)

30 and applied in Tan et al. (2015) but changed such that dense aggregates of same grain size can be obtained. This paper uses the tuned bi-continuos DMRT model as detailed in Zhu et al. (2023). Based on tuned bi-continuous media generation model, 2 show the generated two phased bi-continuous media with grey spots indicating ice and white are represents air. The optical grain size of this generated media is 1mm but because of the aggregation, the ice crystals appear much larger than 1mm. This irregular shaped aggregation of ice crystals causes increased cross-polarized volume backscatter.

[Figure]

**Figure 2.** Bi-continuous Media generation characterized by $< \zeta > = 11000 \text{ m}^{-1}$, $b = 0.4$, and the truncated $\zeta_j = 150 \text{ m}^{-1}$.

35

    Once a volume bi-continuous media has been generated, the next step is to apply Dense Media Radiative Transfer (DMRT) model using full wave simulations of Maxwel's equations to find the scattered fields from the snow volume and then the backscattering coefficients. An overview of DMRT's governing processes is given in Fig. 1. Step 1 consists of a generation of volume bi-continuous media. The second step is to apply the full-wave simulations to compute the compute the scattering,

40 absorption and extinction coefficients, $\kappa_s$, $\kappa_a$ and $\kappa_e$ respectively (Ding et al. (2010)), and phase matrix $\bar{\bar{P}}$ of dense aggregates samples with sample sizes of several wavelengths. Realizations averaging is used to obtain the coherent and incoherent field. Then we solve radiative transfer (RT) equations with calculated phase matrices and extinction coefficients to obtain the backscattering coefficient of a layer of snow pack. The incoherent phase matrix is used in the RT equations.

45     Simulations given in this article consider a single uniform medium layer. For random media with multiple layers, the proposed model is also applicable. Phase matrix and scattering properties are computed for each layer. Then, substitute them into the RT equations to calculate total backscatter of all the layers.

    The ratio of cross polarization can be simply explained by the intuitive relation-

50
$$\frac{\text{cross-pol}}{\text{co-pol}} = \frac{(\text{cross-pol of volume scattering + cross of surface scattering})}{(\text{co-pol of volume scattering + co-pol of surface scattering})}$$

Volume scattering increase with snow depth while surface scattering decreases slightly with snow depth. Thus for the cross-pol/co-pol ratio to increase with snow depth, we need to have cross pol of volume to be larger than cross pol of surface scattering while the co-pol of volume scattering to be less than co-pol of surface scattering.

**3.2 Rough Soil Surface Scattering**

55  The rough soil surface below the snow pack also has a significant impact, particularly at co-pol as rough surface scattering has strong contributions at co-pol even with the attenuation from the snow pack. Therefore, it is important to accurately characterize and calculate $\sigma^{rs}$ from Eq. 1. In this paper, we use a 3D full wave solutions of Maxwell's equations (NMM3D) model to simulate rough surface scattering of both co-polarization and cross-polarization. NMM3D is a method of moments (MoM) based numerical method based on sparse matrix canonical grid (SMCG). In this method, the field interactions on the

60  rough surface is separated into near field and non-near field. The near field is calculated using direct computation of fields. In the non-near field, the sparse impedance matrix is translated over a canonical grid (flat surface in this case) so that the non-near fields are calculated using 2D fast Fourier transform (2DFFT) which makes the computation considerably faster. For each realization, a unique height profile $z = f(x, y)$ from a Gaussian random process is generated based on given rms height, correlation length, and correlation function. Fig. 3 shows the surface profile for the exponential correlation function for 256 ×

65  256 grid points with surface size of $16 \times 16\lambda^2$.

[Figure]

**Figure 3.** Computer generated 3-D profiles for NMM3D simulation (exponential correlation function).

As we are using full wave simulations to find the rough surface scattering of the total backscatter, the cross-polarization component can be accurately calculated. Previous models, although good enough for co-polarized backscatter at smaller rms heights, are not good for calculating cross-polarization. Fundamental method of solving NMM3D is that we calculate the

70  surface electromagnetic currents generated on the rough surface due to the incident radar electric field at a given incidence angle. Once the surface currents are evaluated, the sacttered fields can be calculated easily. Once we have scattered fields and

based on a plane wave incidence, we can calculate the bistatic scattering co-efficient given by-

$$\gamma_{\beta\alpha}(\theta_s, \phi_s, \theta_i, \phi_i) = \lim_{r \to \infty} \frac{4\pi r^2 \mid E^s_\beta \mid^2}{\mid E^i_\alpha \mid^2 A \cos\theta_i} \tag{2}$$

75

Bistatic scattering co-efficient is the in all the upward scattered direction. Backscattered scattering $\sigma_0$ in both co-polarization and cross-polarization is just the bistatic scattering co-efficient going back to the radar in the incidence angle direction. The details for rough surface scattering using NMM3D can be found in Liao et al. (2016).

**3.3 Combined Volume and Surface Scattering Effects**

80

Once we have both volume and surface scattering, we can then calculate contributions from both of the components. For the dense aggregates layer, the cross-polarization to co-polarization ratio of volume scattering is larger than that of rough surface scattering. Thus, although co-polarization volume scattering is less than that of co-polarization surface scattering, the cross-polarizations volume scattering can exceed that of cross-polarization surface scattering. In Fig. 1, we compare volume scattering with surface scattering at 5.4 GHz with respect to two different values of SWE. The random medium has a volume fraction fv of 0.3, a particle size of about 1 mm ($\langle \zeta \rangle$ = 11000 m-1 ), and an aggregation parameter b of 0.4. The permittivity of the ice particles is $\epsilon$p = 3.2 and the background is air. The bottom dielectric surface has a permittivity of $\epsilon$g = 9.0 + 2.5i and an rms height of 1.5 cm. The surface scattering has a small attenuation by the dense aggregates layer. The surface scattering of co-polarization is about -10 dB and the cross-polarization is about -20 dB at 0.55m of SWE. Volume scattering of dense aggregates (b = 0.4) increases with SWE. Note that, for co-polarization, surface scattering is larger than volume scattering until about 0.48 m of SWE. For cross-pol, volume scattering can be larger than surface scattering for snowpacks with a large SWE (above 0.3 m in the example). This is shown in Table 1 where at 0.55 m of SWE, the volume and surface scattering of co-pol are comparable but the volume scattering at cross-pol is 5 dB (3 times) higher than the surface scattering. And as seen in Figure 1, from Section 1, in mountain regions, the SWE constantly goes beyond 0.55m. This is the reason Sentinel-1 sees high sensitivity of cross-pol to snow depth. The same effect is illustrated using snow depths in Figure 11 in Zhu et al. (2023).

85

90

95

|  | SWE (m) | Volume Scattering (dB) | Surface Scattering (dB) |
|---|---|---|---|
| Co-pol | 0.13 | -15.2 | -8.8 |
| VV | 0.55 | -9.4 | -10 |
| Cross-pol | 0.13 | -24.3 | -19.1 |
| VH | 0.55 | -15 | -20.2 |

**Table 1.** Comparison of co-pol and cross-pol backscatter at shallow and very deep snow depths.

**References**

Ding, K.-H., Xu, X., and Tsang, L.: Electromagnetic Scattering by Bicontinuous Random Microstructures With Discrete Permittivities, IEEE Transactions on Geoscience and Remote Sensing, 48, 3139–3151, https://doi.org/10.1109/TGRS.2010.2043953, 2010.

Liao, T.-H., Tsang, L., Huang, S., Niamsuwan, N., Jaruwatanadilok, S., Kim, S.-b., Ren, H., and Chen, K.-L.: Copolarized and Cross-Polarized Backscattering From Random Rough Soil Surfaces From L-Band to Ku-Band Using Numerical Solutions of Maxwell's Equations With Near-Field Precondition, IEEE Transactions on Geoscience and Remote Sensing, 54, 651–662, https://doi.org/10.1109/TGRS.2015.2451671, 2016.

Lievens, H., Demuzere, M., Marshall, H.-P., Reichle, R. H., Brucker, L., Brangers, I., de Rosnay, P., Dumont, M., Girotto, M., Immerzeel, W. W., Jonas, T., Kim, E. J., Koch, I., Marty, C., Saloranta, T., Schöber, J., and De Lannoy, G. J. M.: Snow depth variability in the Northern Hemisphere mountains observed from space, Nature Communications, 10, 4629, https://doi.org/10.1038/s41467-019-12566-y, 2019.

Tan, S., Chang, W., Tsang, L., Lemmetyinen, J., and Proksch, M.: Modeling Both Active and Passive Microwave Remote Sensing of Snow Using Dense Media Radiative Transfer (DMRT) Theory With Multiple Scattering and Backscattering Enhancement, IEEE Journal of Selected Topics in Applied Earth Observations and Remote Sensing, 8, 4418–4430, https://doi.org/10.1109/JSTARS.2015.2469290, 2015.

Tsang, L., Durand, M., Derksen, C., Barros, A. P., Kang, D.-H., Lievens, H., Marshall, H.-P., Zhu, J., Johnson, J., King, J., Lemmetyinen, J., Sandells, M., Rutter, N., Siqueira, P., Nolin, A., Osmanoglu, B., Vuyovich, C., Kim, E., Taylor, D., Merkouriadi, I., Brucker, L., Navari, M., Dumont, M., Kelly, R., Kim, R. S., Liao, T.-H., Borah, F., and Xu, X.: Review article: Global monitoring of snow water equivalent using high-frequency radar remote sensing, The Cryosphere, 16, 3531–3573, https://doi.org/10.5194/tc-16-3531-2022, 2022.

Ulaby, F. T. and Stiles, W. H.: The active and passive microwave response to snow parameters: 2. Water equivalent of dry snow, Journal of Geophysical Research: Oceans, 85, 1045–1049, https://doi.org/https://doi.org/10.1029/JC085iC02p01045, 1980.

Zhu, J., Tsang, L., and Xu, X.: Modeling of Scattering by Dense Random Media Consisting of Particle Clusters With DMRT Bicontinuous, IEEE Transactions on Antennas and Propagation, 71, 3611–3619, https://doi.org/10.1109/TAP.2023.3240562, 2023.